# Advancing Language Multi-Agent Learning with Credit Re-Assignment for Interactive Environment Generalization

**Zhitao He**[1,3*†], **Zijun Liu**[2*], **Peng Li**[1,2‡], **Yi R. (May) Fung**[3], **Ming Yan**[4], **Ji Zhang**[4]
**Fei Huang**[4], **Yang Liu**[1,2‡]
[1]Institute for AI Industry Research (AIR), Tsinghua University, Beijing, China
[2]Dept. of Comp. Sci. & Tech., Institute for AI, Tsinghua University, Beijing, China
[3]Hong Kong University of Science and Technology, Hong Kong, China
[4]Tongyi Lab, Alibaba Group
zhitao.he@connect.ust.hk, zj-liu24@mails.tsinghua.edu.cn

## Abstract

LLM-based agents have made significant advancements in interactive environments, such as mobile operations and web browsing, and other domains beyond computer using. Current multi-agent systems universally excel in performance, compared to single agents, but struggle with generalization across environments due to predefined roles and inadequate strategies for generalizing language agents. The challenge of achieving both strong performance and good generalization has hindered the progress of multi-agent systems for interactive environments. To address these issues, we propose **CollabUIAgents**, a multi-agent reinforcement learning framework with a novel multi-agent credit re-assignment (CR) strategy, *assigning process rewards with LLMs rather than environment-specific rewards and learning with synthesized preference data*, in order to foster generalizable, collaborative behaviors among the role-free agents' policies. Empirical results show that our framework improves both performance and cross-environment generalizability of multi-agent systems. Moreover, our 7B-parameter system achieves results on par with or exceed strong closed-source models, and the LLM that guides the CR. We also provide insights in using granular CR rewards effectively for environment generalization, and accommodating trained LLMs in multi-agent systems. Our work is available at https://github.com/THUNLP-MT/CollabUIAgents.

## 1 Introduction

Autonomous agents have made substantial progress in interactive environments, such as mobile operations and web browsing, by leveraging large language models (LLMs). These agents hold immense potential not only to automate repetitive tasks but also to enhance decision-making and streamline complex workflows. As a result, they can free up human resources for higher-level problem-solving and innovation. The increasing interest in developing such agents is evident in the growing body of work on, for instance, mobile (Rawles et al., 2023; 2025; Zhang et al., 2024b; Deng et al., 2024a; Wang et al., 2024c), web browsing (Shi et al., 2017; Liu et al., 2018a; Yao et al., 2022; Zhou et al., 2024b; Deng et al., 2023; 2024b), and computer using environments (Xie et al., 2024; Sun et al., 2024), and LLM-based agents targeting on these tasks, including single-agent (Yan et al., 2023; Wang et al., 2024b; Hong et al., 2024b; Cheng et al., 2024a; Hu et al., 2024; Zhang et al., 2025) and multi-agent systems (Wang et al., 2024a; Zhou et al., 2023; Zhang et al., 2024c).

However, current efforts in language agent learning still face challenges to balance both performance and generalizability across interactive environments. (1) Single-agent learning

---

*Equal contribution. ‡Corresponding authors: Peng Li, Yang Liu.
†Work done at Institute for AI Industry Research (AIR), Tsinghua University.

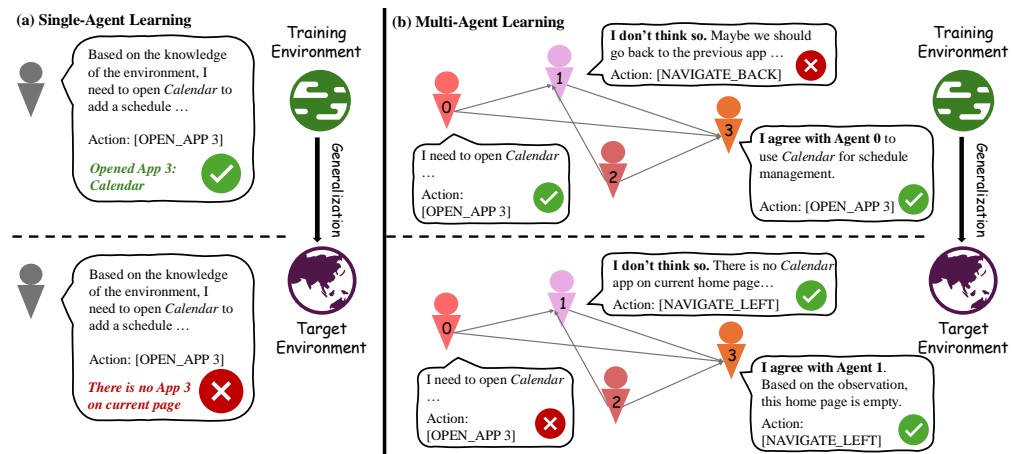

Figure 1: Illustration of (a) **single-agent** and (b) **multi-agent learning** for environment generalization. In single-agent learning, the agent could encounter obstacles when target environments are different from the training one, while in multi-agent learning, collaboration between agents might enable effective decision-making in both environments.

methods (Chen et al., 2023; Gur et al., 2024; Furuta et al., 2024; Bai et al., 2024) heavily relies on in-domain supervised data or rewarding signals to improve environment-specific performance, which restricts its generalization across environments, such as transitioning between web environments using HTML and mobile environments using Android automator. (2) Despite being trained on vast amounts of data from diverse domains, single agents based on open-source LLMs (Zeng et al., 2024; Zhang et al., 2024a; Yin et al., 2024; Liu et al., 2025a) demonstrate only moderate generalization capabilities and lag behind closed-source models. (3) Although multi-agent learning methods (Qiao et al., 2024; Liang et al., 2024) have better performance, existing ones are often constrained by rigid role assignments and lack dedicated designs for generalization, which limits their adaptability to unseen environments, e.g., an agent designed to retrieve documents for question answering is not feasible to handle mobile operations. In short, **it is still unclear how to achieve strong performance and good generalization in interactive environments at the same time**.

In this work, we introduce a reinforcement learning framework for language multi-agent systems, named **CollabUIAgents**, designed to address the challenges above in real-world interactive environments. The methodology is inspired by the success of classic multi-agent reinforcement learning (MARL) to stimulate collaboration (Ma & Luo, 2022). Beyond previous work (Tran et al., 2025), collaboration among langugae agents may also be beneficial for generalization across environments, as illustrated in Figure 1. Compared to existing methods using LLMs in credit assignment for non-language multi-agent settings (Qu et al., 2025; Lin et al., 2025) (Appendix A), further considerations are made to enrich sparse outcome rewards, enhance adaptation across language environments, and enable multi-agent training for language agents. We propose a novel multi-agent credit re-assignment (CR) strategy, *assigning process rewards without using environment-specific outcome rewards, but with the world knowledge embedded in LLMs, and learning with synthesized preference data*, aiming to foster generalizable, collaborative behaviors.

The core of the framework is to **be completely powered by MARL-driven policy learning, rather than fixed role prompts**. Specially, an agentic fine-tuned model, as the *base UIAgent*, is used to initialize a multi-agent system, where each agent has its own policy. After rolling out actions from the policy multi-agent system, the *critic agent* allocates process rewards at both the agent and conversation round levels based on its comprehension of the environment state and agent behaviors. This approach not only enables finer-grained rewards without training numerous value estimators for agents, but also expands data scales by restoring generalizable behaviors from failed trajectories according to the training environment. To avoid misleading the agents with incorrect CR, policies are optimized with preference data synthesized by the *adversarial agent*, to ensure guiding them with correct preference signals. After preference optimization (Rafailov et al., 2023), the multi-agent system is updated in both

model parameters and edges of the communication structure. Empirically, we show that with the CR strategy, (1) preference optimization benefits performance and generalization; (2) edge updates is crucial to orchestrate trained LLMs in multi-agent systems.

CollabUIAgents is capable of cross-environment user interface (UI) interaction, supporting both mobile and web environments. Experimental results demonstrate that the trained multi-agent system achieves superior performance compared to existing agentic learning methods and the strong closed-source model Gemini 1.5 Pro (Gemini Team Google, 2024), with Qwen2 7B (Yang et al., 2024) as the base model. The system also achieves performance comparable to the guidance LLM used in the CR, GPT-4 (OpenAI, 2024), in training environments, and even better in an unseen environment. Especially, CollabUIAgents demonstrates effectiveness in largely gapped generalization from mobile to web environments, still comparable to GPT-4.

In summary, our contributions are as follows:

- We propose a language MARL framework **CollabUIAgents** with a novel CR strategy, to achieve both strong performance and generalization in interactive environments.
- Empirically, we provide insights into the effectiveness of using CR rewards for environment generalization, and the adaptation of trained LLMs in multi-agent systems.
- Extensive experiments show that CollabUIAgents surpasses the performance of strong baselines and shows competitiveness comparable to the guidance LLM of CR in both trained and target environments, even under cross-environment generalization.

## 2 Formulation and Notations

We treat interactive tasks as a sequential decision-making process with single-agent or multi-agent systems in the dynamic environments. Agentic systems make decisions based on the current environment state and accumulated interaction history.

**Task Formulation** Let $S$ be the set of all possible states of a given interactive environment, where each $s \in S$ represents a specific configuration of the UI and hidden states of the environment at a given time step. There is an initial state $s_0$ and a terminal state $s^*$. The action space of an agentic system $\mathcal{G}$ is denoted as $\mathcal{A}$, where $a \in \mathcal{A}$ could represent an action, e.g., clicking buttons, typing, or scrolling through content. The environment evolves according to a transition function $\mathcal{T}(\cdot, \cdot)$:

$$s_{t+1} = \mathcal{T}(s_t, a_t), s_t, s_{t+1} \in S, a_t \in \mathcal{A}, \tag{1}$$

where $s_t$ is the state at time step $t$, and $a_t$ is the action taken by the agent system at that step. The task ends when reaching a terminal state or exceeding the maximum step $T_{\max}$. From the state $s_t$, the observation $o_t$ is derived as formatted descriptions in language. Each agent in the system holds a policy $\pi_i$ and accordingly selects actions based on a shared current observation $o_t$, the history of past interactions $H_{t-1} = (s_0, a_0, ..., s_{t-1}, a_{t-1})$, and the message for agent $\pi_i$ from other agents at conversation round $j$, denoted as $C_t^{i,j}$. Specifically, $C_t^{i,j}$ is omitted for single agents:

$$a_t^{i,j} = \pi_i \left( o_t, H_{t-1}, C_t^{i,j} \right), a_t^{i,j} \in \mathcal{A}, i = 1, ..., |\mathcal{G}|, \tag{2}$$

where $|\mathcal{G}|$ is the number agents in the system. And $a_t$ is determined by an aggregation function $f_{\text{agg}}$ (which is identity for single agents ($|\mathcal{G}| = 1$)):

$$a_t = f_{\text{agg}} \left( \left\{ a_t^{i,j} \middle| i = 1, \cdots, |\mathcal{G}|; j = 1, \cdots, m \right\} \right), \tag{3}$$

where $m$ is the number of conversion rounds. The task goal is to maximize the outcome reward from the environment over a sequence of interactions.

**Interactive Environment** The observation and action space in interactive environments are enormous. Specifically, for the **mobile operation environments**, which offer an interface that allows agents to receive observations and perform actions on mobile devices, the observation space may include high-resolution screenshots and a UI tree from Android automater.

The action space mirrors human interactions, featuring gestures (such as tapping, long-pressing, and swiping), typing, and navigation buttons (e.g., home and back). Complete actions are listed in Table 6. For **web browsing environments**, the observation space may include task description, simplified HTML, and current location. The HTML offers the model both structural and content details of the page, while the current location information allows it to understand its position on the webpage. Consistent with previous work (Lai et al., 2024), we use a unified web browsing action space in both of the aforementioned environments. The actions include hover, select, click, etc. Complete actions are in Table 7.

**Outcome Reward Function** The outcome reward $R_o \in \{0, 1\}$ is defined in the environment based on static rules. Static rules are predefined to check whether agents arrive in a successful terminal state inherent to the given task query. Specifically, in the following experiments, AndriodWorld and MobileMiniWoB++ (Rawles et al., 2025) feature online environments and according state annotations, and Mind2Web (Deng et al., 2023) and AutoWebBench (Lai et al., 2024) use offline trajectories and verifiers. Let the terminal step be $t^*$,

$$R_o = \begin{cases} 1, & \text{if } s_{t^*} = s^* \\ 0, & \text{otherwise} \end{cases}. \tag{4}$$

Thus, the outcome reward is sparse, as only the terminal state $s^*$ gives out positive rewards, posing a challenge to traditional RL approaches.

## 3 CollabUIAgents Framework

The *CollabUIAgents* framework is designed to achieve both high performance and generalizability in interactive environments. It optimizes language multi-agent systems without predefined roles. As shown in Figure 2, the base model undergoes agentic fine-tuning (detailed in Appendix C) and then the MARL process. This section elaborates the multi-agent system architecture and the multi-agent learning process with multi-agent credit re-assignment.

### 3.1 Multi-Agent System Architecture

The architecture of the multi-agent system ($\mathcal{G}$) in CollabUIAgents is consistent with previous work (Zhuge et al., 2024a; Liu et al., 2024), which consists of $|\mathcal{G}| = n$ agents, each represented by a policy $\pi_i$ that communicate with each other through a message network $\mathcal{E}_\mathcal{G}$. As shown in Figure 2, the network is a directed acyclic graph (DAG), where messages are passed from $\pi_{i_1}$ to $\pi_{i_2}$ if there is an edge pointing from $\pi_{i_1}$ to $\pi_{i_2}$. Specifically, the message comes from the output of $\pi_{i_1}$. The framework remains the compatibility for more complex architectures, which is left for future work. We instead use DAGs for simplicity.

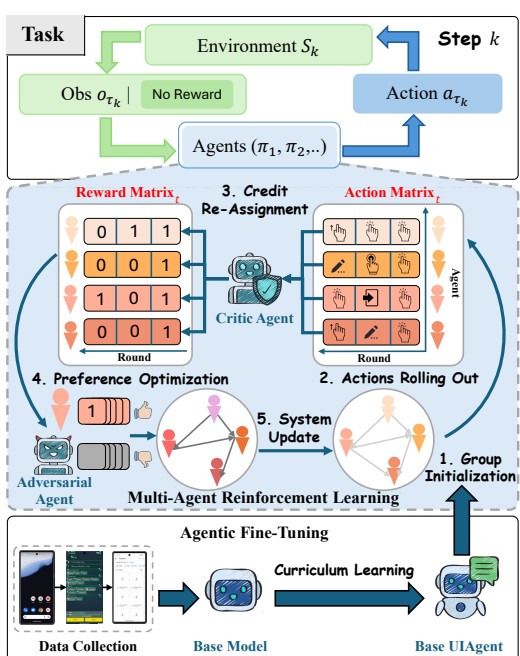

Figure 2: The CollabUIAgents framework with credit re-assignment. The critic agent assess the environment state and the action matrix to get granular rewards. Agents are optimized with synthesized preference data to learn rewarded behaviors.

The agents operate in a topological order, and starting from the source to the sink node, allowing each agent to aggregate all responses from its predecessors to form $C^{i,j}$ in Equation 2. We define the round of conversation as $m$. In each conversation round, all agents output an action and messages *once* along the DAG, and each agent receives outputs from itself at last round besides from predecessors, i.e., we keep a contextual memory

of the size equal to 1. The proper size of contextual memories enhances the diversity of decision making and avoids introducing too long contexts during multi-agent communications. According to Equation 2, at the time step $t$, the system produce an **action matrix** $A_t = (a_t^{i,j}), i = 1, ..., n; j = 1, ..., m$. Then, majority voting is used to decide the final action:

$$a_t = f_{\text{agg}}(A_t) = \text{argmax}_a \sum_{i=1}^{n} \sum_{j=1}^{m} \mathbf{1}_{a_t^{i,j}=a}, \tag{5}$$

where $\mathbf{1}_{\text{condition}}$ is the indicator function. The agents are all required to output an intermediate decision and collaborate towards a common objective, which allows them to function with the same base model for better efficiency, and operate heterogeneously due to different conversation messages.

## 3.2 Language Multi-Agent RL

As shown in Figure 2, the essential stage of *CollabUIAgents* is MARL, after agentic fine-tuning the base model into a *base UIAgent* (Appendix C). We introduce a novel multi-agent **Credit Re-Assignment** (CR) strategy for better MARL. This approach utilizes the *critic agent* to provide fine-grained rewards at both the agent and conversation round levels, enabling agents to learn effectively from general knowledge of LLMs rather than the sparse environment-specfic reward, and improve collaboration by preference optimization with synthesized data from the *adversarial agent*. We also introduce an **Edge Update** trick to better accommodate trained LLMs in the multi-agent system.

**Credit Re-Assignment** Traditional MARL systems use value decomposition for credit assignment (Ma & Luo, 2022), and recent work also uses multiple value estimators to alleviate reward sparcity in interactive environments (Bai et al., 2024). However, training critic models for each agent based on outcome rewards is computationally expensive and may hinder generalization. Instead, using an LLM-based critic agent, represented by $\pi_{\text{critic}}$, we provide process rewards in a finer granularity for each agent in each conversation round. The **action matrix** $A_t$ is assessed by the critic agent based on the observed environment state $o_t$, interaction history $H_{t-1}$, and the task query $q$, generating a **reward matrix** $R_t = (r_t^{i,j})$, $i = 1, ..., n, j = 1, ..., m$, where $r_t^{i,j} \in \{0, 1\}$ denotes the intermediate reward for $a_t^{i,j}$:

$$R_t = \lceil \pi_{\text{critic}} (o_t, H_{t-1}, A_t, q) - 0.5 \rceil, \tag{6}$$

where $\pi_{\text{critic}}(\cdot, \cdot, \cdot, \cdot) \in [0, 1]^{n \times m}$. In our experiments, we use a strong gpt-4o-2024-08-06 to guide the CR process for better validity of the assigned rewards. However, the judgement accuracy would not be perfect when the complexity of the environment increases. To overcome the potential incorrect CR results, we introduce an adversarial agent $\pi_{\text{adv}}$ to synthesize preference data for policy optimization substituting value-based learning. Specially, $\pi_{\text{adv}}$ generates low-quality responses $a_t^{i,j,-}$ paired with $a_t^{i,j}$:

$$a_t^{i,j,-} = \pi_{\text{adv}} \left( o_t, H_{t-1}, C_t^{i,j}, q \right), \text{if } r_t^{i,j} = 1. \tag{7}$$

For better clarity, the adversarial agent is only used for generating inferior actions and responses to help learning, instead of applying attacks (Pinto et al., 2017; Bukharin et al., 2023; Yuan et al., 2023; Lee et al., 2025). Here is the rationale of the design: (1) The critic agent provides a more detailed reward signal for each agent without training multiple value estimators; (2) The critic agent's assessment is based on general pretrained LLMs, which by practice could expand the data scale by restoring intermediate decisions from failed trajectories according to the training environment, and thus may enhance performance; (3) Although errors in CR are inevitable, the synthesized preference data can still provide agents with meaningful preference signals. For example, even if $a_t^{i,j}$ is not the optimal action, it is still better than $a_t^{i,j,-}$. Indeed, environment-specific actions beyond general knowledge of the LLM might be misjudged. We argue that the situation is similar when a value estimator is poorly learned, where the policy might be optimized to a wrong direction, but our approach could keep rolling out new trajectories based on these actions without

unlearning them. The qualitative study shown in Appendix B demonstrates CR results could be valid without the outcome reward.

**MARL with Edge Updates**   Different from classic MARL settings, agents in CollabUIA-gents could communicate and the message network should also be rolled out in the optimization. To alleviate the overhead of learning the optimal network from enormous combinations of edges, we introduce an *edge update* trick, that randomly update edges to form a DAG message network, independently from model parameter updates. Through this process, we encourage agents to learn the awareness of multi-agent collaboration and adapt to diverse message networks rather than being rigid in locally optimal DAG pattern. As shown in Figure 2, the edge update is functioned before rolling out actions from the policy system to coordinate agents in a randomly sampled communication graph. Empirical evidence for its effectiveness is shown in Section 4.3. The overall learning objective for each agent $\pi_i$ is formulated as:

$$
\mathcal{L}_{\mathrm{MARL}}(\pi_i) = -\mathbb{E}_{\mathcal{E}'_{\mathcal{G}} \sim K_{|\mathcal{G}|}} \mathbb{E}_{(s_t, a_t^{i,j}, \hat{H}_t^i) \sim \mathcal{P}(\mathcal{G}, \mathcal{E}'_{\mathcal{G}})}
$$
$$
\sum_{t=0}^{T_{\max}} \sum_{j=1}^{m} \left[ \log \sigma \left( \beta \left( \frac{\log \pi_{\theta_i}(a_t^{i,j}|o_t, \hat{H}_t^i)}{\log \pi_{\mathrm{ref}_i}(a_t^{i,j}|o_t, \hat{H}_t^i)} - \frac{\log \pi_{\theta_i}(a_t^{i,j,-}|o_t, \hat{H}_t^i)}{\log \pi_{\mathrm{ref}_i}(a_t^{i,j,-}|o_t, \hat{H}_t^i)} \right) \right) \right] \cdot \mathbf{1}_{r_t^{i,j}=1}, \quad (8)
$$

where $\hat{H}_t^i = \{H_{t-1}, C_t^{i,j}, q\}$, $\theta_i$ are the updating parameters of agent $\pi_i$ and $\mathrm{ref}_i$ is the original model used as reference policy, $K_{|\mathcal{G}|}$ is a fully connected graph of $|\mathcal{G}|$ nodes, $\mathcal{E}'_{\mathcal{G}}$ represents a DAG subgraph *randomly* sampled from $K_{|\mathcal{G}|}$, $\mathcal{P}(\mathcal{G}, \mathcal{E}'_{\mathcal{G}})$ is the preference dataset sampled with agents in the message network $\mathcal{E}'_{\mathcal{G}}$, $\sigma$ is the sigmoid function, $\beta$ is the hyper-parameter, and $\pi_\theta, \pi_{\mathrm{ref}}$ are the base model and reference model. This objective encourages the policy $\pi_i$ to assign higher probabilities to preferred actions $a_t^{i,j}$ compared to adversarial actions $a_t^{i,j,-}$. The policy is updated online and off-policy, similar to previous work (Bai et al., 2024).

### 3.3   Cross-Environment Adaptation

One of the key strengths of the *CollabUIAgents* framework is its ability to generalize across different interactive environments, such as from mobile operations to web browsing environments. The framework supports two approaches for adaptation.

**Direct Transfer**   In scenarios where the new environment shares similarities with the training environment, agents can be deployed directly without additional training. For example, agents trained in mobile UI environments can directly apply their knowledge to web environments, leveraging the knowledge of common interaction patterns and UI elements. The multi-agent setup and according MARL stage are keys to decrease error rates through enhancing generalizable collaborative behaviors in agents as expected. The effectiveness is shown in Section 4.2 (applying CollabUIAgents$_{\mathrm{mobile}}$ to web environments).

**Continual MARL**   When the new environment presents significant differences or the highest success rates are required, agents can undergo further training using MARL with the CR strategy in new environments. This continual learning approach allows agents to refine their policies without stashing the knowledge of previous environments, showing substantial performance increase without re-trainig the agent system (Section 4.2, CollabUIAgents$_{\mathrm{m}\to\mathrm{web}}$).

## 4   Experiment

### 4.1   Experimental Settings

**Environments**   We conduct experiments in both mobile operation and web browsing environments. For the mobile environments, we use AndroidWorld (Rawles et al., 2025) for training and MobileMiniWoB++ (Rawles et al., 2025) for testing: (1) **AndroidWorld** has 116 programmatic tasks across 20 real-world apps, such as Chrome, Markor, and Pro Expense. (2) **MobileMiniWoB++** is derived from MiniWoB++ (Shi et al., 2017), which is a web-based

| Method | #Params/#Agents | Input | $SR_{AndroidWorld}$ | $SR_{MMiniWoB++}$ | $\Delta_{Generalization}$ |
|---|---|---|---|---|---|
| *Agents based on Closed-Source LLMs* | | | | | |
| M3A (GPT-4) | N/A | Text | **30.6** | 59.7 | - |
| M3A (GPT-4) | N/A | Multimodal | 25.4 | **67.7** | - |
| SeeAct (GPT-4) | N/A | Multimodal | 15.5 | 66.1 | - |
| M3A (Gemini 1.5 Pro) | N/A | Text | 19.4 | 57.4 | - |
| M3A (Gemini 1.5 Pro) | N/A | Multimodal | 22.8 | 40.3 | - |
| *Agents based on Open-Source LLMs* | | | | | |
| Qwen2 | 7B/1A | Text | 6.2 | 12.9 | - |
| Qwen2 | 7B/4A | Text | 4.2 | 15.2 | - |
| Qwen2 VL | 2B/1A | Multimodal | 0.0 | 10.0 | - |
| Qwen2 VL | 2B/4A | Multimodal | 0.0 | 11.5 | - |
| Qwen2.5 VL | 3B/1A | Multimodal | 10.1 | 18.7 | - |
| Qwen2.5 VL | 3B/4A | Multimodal | 12.6 | 21.3 | - |
| InfiGUIAgent (Qwen2 VL) | 2B/1A | Multimodal | 9.1 | 15.6 | 5.6 |
| DigiRL (Qwen2.5 VL) | 3B/1A | Multimodal | 22.3 | 35.2 | 16.5 |
| DigiRL (Qwen2.5 VL) | 3B/4A | Multimodal | 20.1 | 38.7 | 20.0 |
| **Our Methods** | | | | | |
| UIAgent (Qwen2) | 7B/1A | Text | 18.9 | 48.4 | 35.5 |
| UIAgent (Qwen2) | 7B/4A | Text | 21.4 | 53.2 | 40.3 |
| UIAgent (Qwen2) | 7B/6A | Text | 18.9 | 54.3 | 41.5 |
| CollabUIAgents$_{mobile}$ (Qwen2) | 7B/4A | Text | 29.3 | **61.2** | **48.3** |
| CollabUIAgents$_{mobile}$ (Qwen2) | 7B/6A | Text | **32.7** | 59.7 | 46.9 |

Table 1: Experimental results on mobile operation environments. Success rates (SR) in AndoridWorld and MobileMiniWoB++ (MMiniWoB++) are listed. $\Delta_{Generalization}$ indicates the performance gap between the base model and agent learning methods based on the model in MobileMiniWoB++. "7B/4A" denotes a four-agent system upon a 7B model.

benchmark. MobileMiniWoB++ shares the same observation space as AndroidWorld and supports 92 tasks from MiniWoB++. We use the success rate (SR) as an evaluation metric. For the web environments, we leverage Mind2Web (Deng et al., 2023) for training and AutoWebBench (Lai et al., 2024) for testing: (1) **Mind2Web** features over 2,000 open-ended tasks sourced from 137 websites in 31 different domains. (2) **AutoWebBench** is a bilingual benchmark featuring approximately 10,000 traces, from mainstream Chinese and English websites, providing a diverse dataset for web browsing. We use the step-success rate (SSR) as the evaluation metric. For agent learning methods on open-source LLMs, we use the performance gap between the base model and the trained model in unseen environments, $\Delta_{Generalization}$, to indicate generalizability for each agent learning method.

**Evaluated Methods** We compare our framework against the following methods under their original settings: (1) **M3A** (Rawles et al., 2023) is a prompt-based multimodal agent, which combines ReAct- (Yao et al., 2023) and Reflexion-style (Shinn et al., 2023) prompting to interpret user instructions and screen content, then update its decisions. (2) **SeeAct** (Zheng et al., 2024) is a prompt-based navigation agent originally designed for GPT-4V to perform actions with visual input and textual choices. (3) **InfiGUIAgent** (Liu et al., 2025a) fine-tunes a generalist mobile operator model with multimodal input. (4) **DigiRL** (Bai et al., 2024) is an off-policy RL algorithm for single agents trained on the same dataset as CollabUIAgents, based on Qwen2.5 VL 3B (Wang et al., 2024d). (5) **SeeClick** (Cheng et al., 2024b) is a fine-tuned visual GUI agent that automates tasks relying on screenshots and employs GUI grounding. We leverage Qwen2 7B as our base model and evaluate the following systems derived from the model: (1) **Base Model** directly calls the general instruction-tuned model to interact with the environment without fine-tuning. (2) **Base UIAgent** is the base model that has undergone agentic fine-tuning in AndroidWorld. (3) **CollabUIAgents$_{mobile}$** is our framework trained on AndroidWorld with $n = 4, m = 3$. (4) **CollabUIAgents$_{m \rightarrow web}$** builds upon CollabUIAgents$_{mobile}$ with continue MARL on the training set of Mind2Web, which is autonomously collected with the pipeline in Appendix C and D.

## 4.2 Main Results

**Effectiveness in Mobile Environments** In this section, we explore the effectiveness of our proposed method in trained and unseen mobile operation environments. Experimental results in mobile environments are shown in Table 1. The performance of proprietary

| Method | #Params/#Agents | Input | SSR$_{Mind2Web}$ | SSR$_{AutoWebBench}$ | $\Delta_{Generalization}$ |
|---|---|---|---|---|---|
| *Agents based on Closed-Source LLMs* | | | | | |
| GPT-3.5-Turbo | N/A | Text | 17.4 | 10.7 | - |
| GPT-4 | N/A | Text | **30.9** | **37.8** | - |
| Claude2 | N/A | Text | - | 10.5 | - |
| *Agents based on Open-Source LLMs* | | | | | |
| Qwen2 | 7B/1A | Text | 7.4 | 8.5 | - |
| LLaMA2 | 7B/1A | Text | - | 2.9 | - |
| LLaMA2 | 70B/1A | Text | - | 10.6 | - |
| SFT (Qwen VL) | 9.6B/1A | Multimodal | 10.1 | - | - |
| SeeClick (Qwen VL) | 9.6B/1A | Multimodal | 20.9 | - | - |
| **Our Methods** | | | | | |
| UIAgent (Qwen2) | 7B/1A | Text | 11.9 | 12.8 | 4.3 |
| UIAgent (Qwen2) | 7B/4A | Text | 13.2 | 14.0 | 5.5 |
| CollabUIAgents$_{mobile}$ (Qwen2) | 7B/4A | Text | 16.2 | 17.7 | 9.2 |
| CollabUIAgents$_{m \to web}$ (Qwen2) | 7B/4A | Text | **30.7** | **34.7** | **26.2** |

Table 2: Experimental results on web browsing environments. Average step success rates (SSR) in Mind2Web and AutoWebBench are reported. "SFT" denotes the base model is supervised fine-tuned in Mind2Web. $\Delta_{Generalization}$ indicates the gap between the base model and agent learning methods based on the model in AutoWebBench. "7B/4A" denotes a four-agent system upon a 7B-parameter model.

models face instability with input format, e.g., text-only or multi-modal input. With appropriate agentic fine-tuning, open-source models like Qwen2 7B could gain significant performance improvement, and even higher within multi-agent systems ("Base UIAgent"). CollabUIAgents$_{mobile}$ achieves the best results among systems based on open-source LLMs. Remarkably, it outperforms Gemini 1.5 Pro in both environments and achieves performance comparable to or better than GPT-4, due to its instability. These outcomes demonstrate the effectiveness of our framework and provide evidence for the validity of the CR strategy. On the other hand, InfiGUIAgent uses a large amount of training data but falls behind other methods; DigiRL shows good performance in the training environment, but falls short in generalization compared to CollabUIAgents$_{mobile}$, which improves greatly on unseen tasks. In addition, increasing the number of agents leads to further improvement in the training environment but does not improve as much in unseen environments, showing a trade-off. This identifies the importance of maintaining the same number of agents in both the MARL and the deployment stages for the best effectiveness and generalization.

> **Takeaway 1:** The critic agent could generate valid process rewards to guide the policy multi-agent system learning during CR.

**Generalization from Mobile to Web Environments**     In this section, we examine the cross-environment generalization capabilities of our proposed method. Results for web environments are presented in Table 2. Directly transferring CollabUIAgents$_{mobile}$ obtained from the AndroidWorld environment yields substantial performance improvement over Base UIAgent or vanilla Qwen2; however, the absolute gains remain modest and slightly lags behind fine-tuned SeeClick in the training environment. CollabUIAgents$_{mobile}$ with continue MARL on data from Mind2Web, collected with the pipeline depicted in Appendix C, could significantly improve the generalization on web browsing environments. CollabUIAgents$_{m \to web}$ achieves results comparable to GPT-4, without training on large scaled web data. It is also noteworthy that we do not require human-annotated data for the Mind2Web environment, which is a significant advantage in transferring the agent system to new environments. Also, generalization performance ($\Delta_{Generalization}$) in the unseen environment indicates that our method generalizes well in diverse web browsing tasks.

> **Takeaway 2:** Continue MARL significantly improves the language multi-agent system for cross-environment generalization.

## 4.3 Ablation Study

The results of the ablation study are presented in Table 3, including replacing preference optimization with rejective fine-tuning, removing the CR, and removing edge updates in MARL. The empirical findings highlight the following key insights:

(1) Further training of the Base UIAgent with trajectory data using either rejective SFT ("CollabUIAgents$_{mobile}$ w/ PO $\rightarrow$ RFT") or DPO on whole trajectories ("CollabUIAgents$_{mobile}$ w/o CR") improves performance, with DPO showing superior results. The primary distinction between these methods is that SFT can only learn from correct actions, while DPO can learn from both correct and incorrect actions. (2) CollabUIAgents$_{mobile}$ introduces credit re-assignment, providing more granular feedback that facilitates exploration of the large action space at each step. The synthesized preference data also helps generalization through preference optimization, compared to CollabUIAgents$_{mobile}$ w/o CR which only uses the outcome reward $R_o$. This boosts both performance and generalizability, yielding the best overall results.

> **Takeaway 3:** Preference optimization with synthesized data enhances performance and generalization with CR-generated rewards.

(3) Combining multiple agents based on a vanilla base model using random edges leads to modest improvements ("Base UIAgent" ($n = 4$)), and the similar trend exists for DigiRL in Table 1, underscoring the importance of proper agentic fine-tuning and MARL with online multi-agent trajectories. (4) A comparison between systems with and without edge updates ("CollabUIAgents$_{mobile}$" vs. "w/o edge update") demonstrates that the edge update trick contributes to further accommodating language agents in complex multi-agent systems with communications.

| Method | SR$_{\textbf{AndroidWorld}}$ | SR$_{\textbf{MMiniWoB++}}$ |
|---|---|---|
| *Single-Agent Systems* | | |
| Qwen2 | 6.2 | 12.9 |
| UIAgent (LLaMA2) | 15.1 | 43.7 |
| UIAgent (Qwen2) | 18.9 | 48.4 |
| UIAgent$_{self-critic}$ (Qwen2) | 10.7 | 19.5 |
| *Multi-Agent Systems ($n = 4$)* | | |
| Qwen2 | 8.6 | 16.1 |
| UIAgent (Qwen2) | 21.4 | 53.2 |
| UIAgent$_{self-critic}$ (Qwen2) | 12.5 | 26.1 |
| CollabUIAgents$_{mobile}$ | **29.3** | **61.2** |
|   w/ PO $\rightarrow$ RFT | 23.2 | 54.8 |
|   w/o CR | 25.0 | 56.4 |
|   w/o Edge Update | 27.6 | 58.1 |
| CollabUIAgents$_{m\rightarrow web}$ | 26.7 | 58.1 |

Table 3: Ablation study. Success Rates (SR) in the AndroidWorld and MobileMiniWoB++ (MMiniWoB++) environments are reported. All methods are based on Qwen2 7B. "PO" denotes preference optimization, and "PO $\rightarrow$ RFT" means performing rejective fine-tuning based on CR rewards, i.e., filtering unrewarded data.

> **Takeaway 4:** Edge updates during MARL help accommodate language agents in the multi-agent system.

(5) After cross-environment reinforcement learning on the web, CollabUIAgents$_{m\rightarrow web}$ exhibits impressive autonomous adaptability in the new environment, with only minor performance fluctuations in the original mobile environment, thereby validating the stability of our method.

## 5 Related Work

**Agents on Interactive Environments** Before the advent of LLMs, agents relied on traditional RL to perform interactions such as clicking and typing (Liu et al., 2018b; Humphreys et al., 2022). However, recent advancements have shifted towards leveraging foundation

models with in-context learning or fine-tuning across various interfaces, including mobile (Wang et al., 2023; Hong et al., 2024b), web (Lai et al., 2024; Deng et al., 2024b), and computer using environments (Xu et al., 2024a; Wu et al., 2024c). Recently, there are emerging methods designing process rewards (He et al., 2024a; Pan et al., 2024; Xu et al., 2024b; Wu et al., 2024b; He et al., 2025; Xu et al., 2025; Liu et al., 2025b), synthetic data (Yuan et al., 2024; Qin et al., 2025) and language RL (Bai et al., 2024) for better performing single agents.

**Interactive Environments for Agents** To effectively evaluate language agents, it is essential to create environments that replicate real-world conditions and deliver accurate rewards (Rawles et al., 2023; Deng et al., 2023). MiniWoB++ (Shi et al., 2017) is a lightweight framework that features small, synthetic HTML pages with parameterized tasks. WebArena (Zhou et al., 2024a) and its visual counterpart, VisualWebArena (Koh et al., 2024), simulate websites spanning up to distinct domains, while WorkArena (Drouin et al., 2024) focuses on enterprise software. For more specialized environments, WebShop (Yao et al., 2022) simulates an e-commerce platform for online shopping.

**Prompt-Based Multi-agent Learning** Collaboration among multiple LLM agents has shown effective for various tasks (He et al., 2023; Hong et al., 2024a; Wu et al., 2024a; He et al., 2024c; Jin et al., 2024; Qian et al., 2024; He et al., 2024b; Wang et al., 2025). However, employing a static architecture without team optimization may restrict the performance and generalization. Chen et al. (2024) selects a fixed number of agents from a set of manual prompt candidates via an additional LLM during each round of discussion. Zhuge et al. (2024b) unify language agent systems by describing them as optimizable computational graphs and develop optimization methods for nodes and edges, enabling automatic improvements of agent prompts and inter-agent orchestration.

# 6 Conclusion

In this paper, we introduce CollabUIAgents, a novel multi-agent reinforcement learning framework aimed at addressing the challenge of balancing strong performance and generalization in interactive environments. The framework employs a credit re-assignment (CR) strategy that utilizes world knowledge embedded in LLMs to assign process rewards, and optimize policies with synthesized preference data. Through extensive experimentation, the proposed framework not only surpasses existing methods in terms of performance metrics but also exhibits exceptional generalization capabilities. Notably, it achieves results that are comparable to, and in some cases even exceed, those of closed-source models when deployed in previously unseen environments. Overall, CollabUIAgents presents a promising solution to the limitations of current agent learning methods by offering a more flexible, data-efficient, and generalizable approach for real-world applications.

# Acknowledgments

This work is supported by the National Natural Science Foundation of China (No. 62276152), and funding from Wuxi Research Institute of Applied Technologies, Tsinghua University under Grant 20242001120.

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

## A Comparisons with Non-Language Multi-Agent Learning Methods

To situate CollabUIAgents among recent studies that also employ LLMs for multi-agent credit assignment, we contrast it with the most closely related approaches of Lin et al. (2025); Qu et al. (2025). Table 4 summarizes the key differences.

| Feature | CollabUIAgents | LCA (Lin et al., 2025) | LaRe (Qu et al., 2025) |
|---|---|---|---|
| Primary focus | Perf. & gen. of *language* agents across interactive environments | Sparse-team credit assignment | Episodic credit assignment (redundancy/ambiguity) |
| Core application | Mobile/web dependent language tasks | GridWorld, Pistonball | MuJoCo, MPE |
| Assigned reward | Fine-grained *process* rewards & synthesized DPO preference | Dense potential-based rewards from LLM rankings | Proxy rewards via latent encoder–decoder |
| Generalization goal | ✓ | ✗ | ✗ |
| Interactive env. | ✓ | ✗ | ✗ |
| Dynamic MARL communication | ✓ | ✗ | ✗ |

Table 4: Comparison with other LLM-based credit-assignment methods for non-language agents.

- **Specifically Designed for Language Agents & Interactive Environments:** Unlike LCA and LaRe—both devised for classical MARL control domains—our framework is purpose-built for language agents acting in complex UI environments (e.g. mobile navigation, web browsing) that combine large observation spaces with natural-language actions and agent communication.

- **LLM-based *process* reward assignment:** We innovatively employ an LLM as a Critic Agent to assign fine-grained "process rewards". The assigned step-level and agent-level process rewards reflect the semantic utility of each action; these rewards are then distilled into synthesized preference data for DPO, moving beyond sparse environmental signals.

- **Emphasis on Cross-Environment Generalization:** Our CR strategy is designed to leverage the general world knowledge of LLMs. Combined with continual MARL and mechanisms like communication structure (edge) updates, this significantly enhances the generalization capabilities of language multi-agent systems in unseen environments.

This targeted design delivers superior performance and generalisation for language-driven multi-agent systems, establishing CollabUIAgents as a distinct advancement over contemporaneous credit-assignment methods.

## B Case Studies on Mobile Operation Environments

Figure 3 shows an example of task execution steps in the AndroidWorld environment during the rolling out phase, where agents in the multi-agent system collectively accomplish the task within the shortest path. We also demonstrate that the reward from the CR process is correctly identified for each action (only a part of rolled out actions are shown).

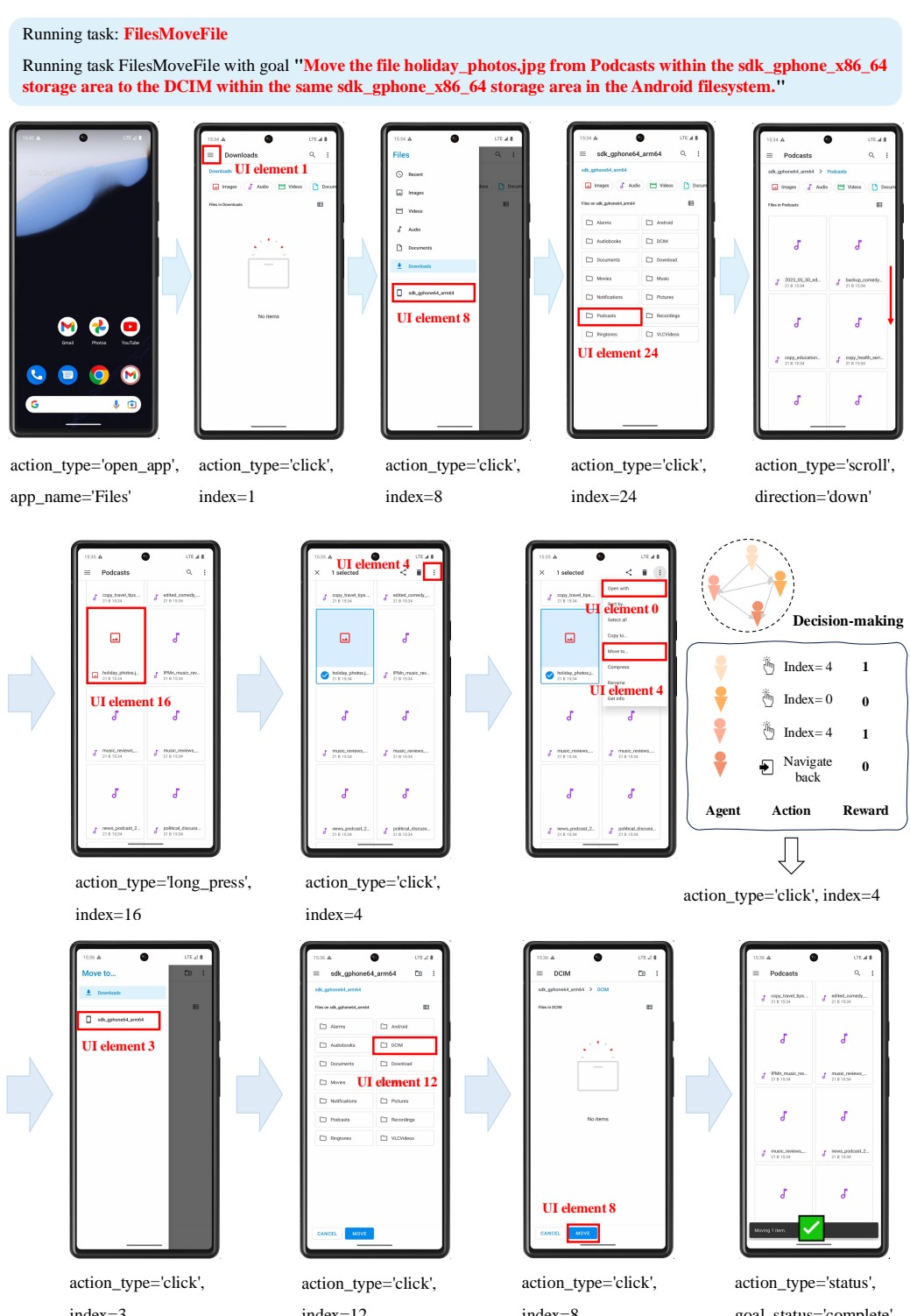

Figure 3: An example of task execution steps of CollabUIAgents_mobile which is trained on Qwen2 7B and facilitates 4 agents engaging in 3 rounds of conversation. "action_type" represents the action taken, and "index" represents the index of the UI element. The positions of the relevant elements on the UI interface are marked in red.

# C   Agentic Fine-Tuning in CollabUIAgents

## C.1   Methodology

The agentic fine-tuning process of the CollabUIAgents framework focuses on adapting base models to new environments through curriculum-based single-agent training (Bengio et al., 2009). The training data is synthesized automatically with a multi-agent data synthesis pipeline and consists of progressively complex instruction sets in three levels, designed to help agents build a strong foundation of environmental knowledge. The UI agent generates responses to synthesize queries faithfully, the adversarial agent generates negative samples, and the critic agent grades process rewards.

**Curriculum Structure**   The training data is divided into three categories, as collected in Figure 4:

(1) **Basic Environmental Knowledge**: This data segment includes identifying UI elements and understanding their properties. We categorize basic knowledge into two types: **UI Understanding** (coarse-grained): This refers to a broad understanding of the layout and information contained in the UI, such as identifying the purpose of the interface. **UI Element Recognition** (fine-grained): Since UI typically contains a large number of densely packed interface, the agent needs to be able to distinguish between different types of elements, such as buttons, input fields, and drop-down menus, and understand the associated actions. We develop a series of queries accordingly in Appendix D.4.1, and randomly select UI elements and the layout to assemble queries for the UI Agent.

(2) **Simple Instruction Knowledge**: The agents are tasked with performing basic interactions, such as clicking or typing, in response to simple instructions. Specifically, given the complete action space, we prompt the UI agent to generate possible instructions related to a random UI element, and their corresponding responses. For example, in Figure 4, the UIAgent was prompted to generate an instruction for element 9 ("*selecting the M4a format*") and then generates the corresponding response to interact with it. By learning this type of knowledge, the agent lays the foundation for completing a complex sequential decision-making process.

(3) **Process Preference Knowledge**: Real-world interactive tasks is quite difficult, and even the most advanced large language model, GPT-4, shows a low task completion rate (30%) in the mobile environment Android-World (Rawles et al., 2025). Training a model solely on scarce successful trajectories still inevitably results in errors. Therefore, as illustrated below Figure 4, we introduce the adversarial agent against the UI agent, and the

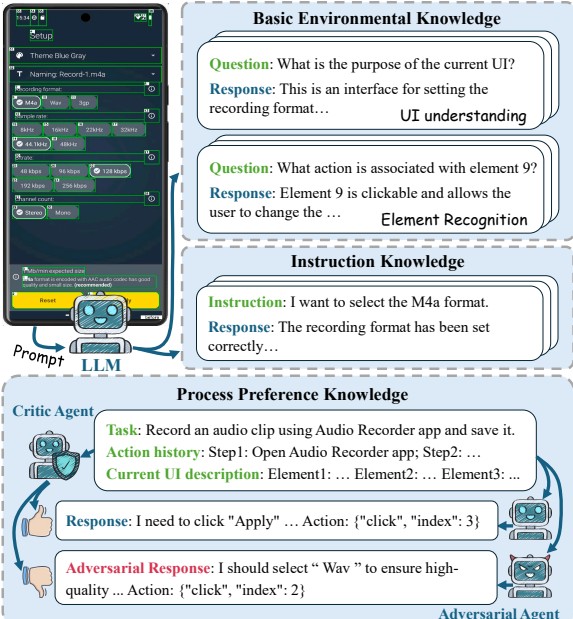

Figure 4: Our multi-agent autonomous data synthesis pipeline. Given a task, the pipeline can autonomously collect data from each step covering basic environmental knowledge, simple instruction knowledge, and process preference knowledge in interactive environments.

critic agent to score all actions, obtaining process preference data with step-level rewards. By learning from process preference data, the agent can better distinguish between correct and incorrect actions during the process, ultimately improving task completion rates. The distribution of the collected data can be found in Appendix D.4.2.

The base model is first trained using supervised fine-tuning (SFT) on the basic environmental knowledge and the simple instruction knowledge, progressively. The learning objective is:

$$\mathcal{L}_{\text{SFT}} = -\mathbb{E}_{(s,a)\sim\mathcal{D}}\left[\log \pi_\theta(a|s)\right], \tag{9}$$

where $\mathcal{D}$ represents the dataset of state-action pairs. Following SFT, the base model are further optimized using direct preference optimization (DPO) on the process preference knowledge:

$$\mathcal{L}_{\text{DPO}} = -\mathbb{E}_{(s,a^+,a^-)\sim\mathcal{P}}\left[\log \sigma\left(\beta\log\frac{\pi_\theta(a^-|s)}{\pi_{\text{ref}}(a^-|s)} -\beta\log\frac{\pi_\theta(a^+|s)}{\pi_{\text{ref}}(a^+|s)}\right)\right], \tag{10}$$

where $\mathcal{P}$ is the preference-labeled dataset, $a^+, a^-$ denote positive and adversarial actions, $\sigma$ is the sigmoid function, $\beta$ is the hyper-parameter, and $\pi_\theta, \pi_{\text{ref}}$ are the base model and reference model.

### C.2 Ablation Study of Agentic Fine-tuning

In this stage, we develop an automated data synthesis method to gather basic environmental knowledge, simple instruction knowledge, and process preference knowledge from the dynamic mobile environment, AndroidWorld.

| Method | $\text{SR}_{\text{AndroidWorld}}$ | $\text{SR}_{\text{MMiniWoB++}}$ |
|---|---|---|
| Qwen2 | 6.2 | 12.9 |
| + Basic knowledge SFT | 12.1 | 22.5 |
| + Instruction SFT | 15.1 | 35.8 |
| + Process DPO | **18.9** | **48.4** |

Table 5: Ablation study of the agentic fine-tuning process on mobile operation environments. Success Rates (SR) in the AndroidWorld and MobileMiniWoB++ (MMiniWoB++) environments are reported. All methods are based on Qwen2 7B.

Based on the upper section of Table 5, we derive the following conclusions: (1) Incorporating basic environmental knowledge data substantially improves the base model's comprehension of dynamic mobile environments, achieving a absolute performance gain of 5.9% in AndroidWorld and 9.6% in MobileMiniWoB++ ("+ Basic knowledge SFT"). It is noteworthy that the collected UI page information excludes app-specific details of MobileMiniWoB++, yet training with general knowledge from AndroidWorld enables the model to generalize effectively to new apps and tasks. (2) Simple instruction knowledge data is crafted to guide the agent in interacting with the environment using actions from the specified action space. Our experiments demonstrate that incorporating instruction data further enhances the base model's ability to complete simple tasks within UI environments ("+ Instruction SFT"). (3) A key advantage of our proposed method is its ability to learn from incorrect actions using process preference knowledge data. Experimental results confirm that this addition significantly boosts performance ("+ Process DPO"). The improvement is more pronounced in the MobileMiniWoB++ environment, which we attribute to the simplicity of its tasks. Fewer steps are required to complete these tasks, leading to greater performance gains.

## D Experiment Details

### D.1 Action Space in the Environments

Table 6 and 7 show the action spaces of agents in mobile and web environments, respectively.

### D.2 Detailed Results in Web Browsing Environments

Detailed experimental results in web browsing environments are shown in Table 8 and Table 9, corresponding to the results in Table 2.

| Action | Description |
|---|---|
| CLICK | Tap once on the element |
| DOUBLE_TAP | Quickly tap the element twice |
| SCROLL | Slide the screen to view more content |
| SWIPE | Quick swipe across the screen |
| INPUT_TEXT | Type text into the element |
| NAVIGATE_HOME | Return to the home screen |
| NAVIGATE_BACK | Go back to the previous screen |
| KEYBOARD_ENTER | Press the enter key |
| OPEN_APP | Launch an app |
| STATUS | Check task status |
| WAIT | Pause briefly |
| LONG_PRESS | Tap and hold on the element |
| ANSWER | Give a response |
| UNKNOWN | Undefined action |

Table 6: The action space in mobile environment.

| Action | Description |
|---|---|
| CLICK | Click at an element |
| HOVER | Hover on an element |
| SELECT | Select option in an element |
| TYPE_STRING | Type to an element |
| SCROLL_PAGE | Scroll up or down of the page |
| GO | Go forward or backward of the page |
| JUMP_TO | Jump to URL |
| SWITCH_TAB | Switch to i-th tab |
| USER_INPUT | Notify user to interact |
| FINISH | Stop with answer |

Table 7: The action space in web environment.

## D.3 Method Implementation

### D.3.1 Baselines

(1) **M3A** (Rawles et al., 2023) is a prompt-based multimodal agent, which combines ReAct- (Yao et al., 2023) and Reflexion-style (Shinn et al., 2023) prompting to interpret user instructions and screen content, then update its decisions. (2) **SeeAct** (Zheng et al., 2024) is a prompt-based navigation agent originally designed for GPT-4V to perform actions with visual input and textual choices. (3) **InfiGUIAgent** (Liu et al., 2025a) is used with the temperature set to 0.1, and other settings remain default. (4) **DigiRL** (Bai et al., 2024) is reproduced on the same dataset as CollabUIAgents, based on Qwen-2.5-VL-3B-Instruct (Wang et al., 2024d). Due to limited computational resource and the high costs of RL training with multiple value estimators, a larger scaled base model could not be applied. (5) **SeeClick** (Cheng et al., 2024b) is a fine-tuned visual GUI agent that automates tasks relying on screenshots and employs GUI grounding. For those baselines based on closed-source LLMs, we conduct experiments using prompts and settings consistent with Rawles et al. (2025), with model data up to August 2024.

### D.3.2 CollabUIAgents Framework

In the agentic fine-tuning process for data collection, we employ GPT-4o-2024-08-06 (OpenAI, 2024) as the UI agent. We set the temperature of different agents to 0.1 to ensure the accuracy of their responses. For the backbone model in the policy multi-agent system, we utilize Qwen2 7B (Yang et al., 2024). Detailed process is depicted in Appendix D.4. In this phase, the model undergoes both supervised learning and off-line preference learning with DPO. Details of the experimental data statistics can be found in Appendix D.4.2. During the supervised fine-tuning phase, the model's context length is set to 8,192, with a learning rate

| System | #Params/#Agents | Input | Cross-Task | Cross-Website | Cross-Domain | Avg. |
|---|---|---|---|---|---|---|
| *Agents based on Closed-Source LLMs* | | | | | | |
| GPT-3.5-Turbo | N/A | Text | 17.4 | 16.2 | 18.6 | 17.4 |
| GPT-4 | N/A | Text | **36.2** | **30.1** | **26.4** | **30.9** |
| *Agents based on Open-Source LLMs* | | | | | | |
| Qwen-VL* | 9.6B/1A | Multimodal | 12.6 | 10.1 | 8.0 | 10.2 |
| SeeClick* | 9.6B/1A | Multimodal | 23.7 | 18.8 | 20.2 | 20.9 |
| Qwen2 | 7B/1A | Text | 8.6 | 6.3 | 7.5 | 7.4 |
| **Our Methods** | | | | | | |
| Base UIAgent | 7B/1A | Text | 13.4 | 10.6 | 11.8 | 11.9 |
| Base UIAgent | 7B/4A | Text | 15.7 | 11.2 | 12.9 | 13.2 |
| CollabUIAgents$_{mobile}$ | 7B/4A | Text | 19.2 | 13.8 | 15.5 | 16.2 |
| CollabUIAgents$_{m \to web}$ * | 7B/4A | Text | **34.5** | **32.7** | **25.1** | **30.7** |

Table 8: Step Success Rates (SSR) in the Mind2Web environment. * indicates the system fine-tunes its base model on the corresponding training set of the environment.

| System | #Params/#Agents | English | | Chinese | | Avg. |
|---|---|---|---|---|---|---|
| | | Cross-Task | Cross-Domain | Cross-Task | Cross-Domain | |
| *Agents based on Closed-Source LLMs* | | | | | | |
| GPT-3.5-Turbo | N/A | 12.1 | 6.4 | 13.5 | 10.8 | 10.7 |
| GPT-4 | N/A | **38.6** | **39.7** | **36.7** | **36.3** | **37.8** |
| Claude2 | N/A | 13.2 | 8.1 | 13.0 | 7.9 | 10.5 |
| *Agents based on Open-Source LLMs* | | | | | | |
| LLaMA2 | 7B/1A | 3.3 | 2.5 | - | - | 2.9 |
| LLaMA2 | 70B/1A | 8.3 | 8.9 | - | - | 10.6 |
| Qwen2 | 7B/1A | 8.6 | 9.4 | 8.1 | 7.8 | 8.5 |
| **Our Methods** | | | | | | |
| Base UIAgent | 7B/1A | 12.0 | 13.3 | 12.7 | 13.4 | 12.8 |
| Base UIAgent | 7B/4A | 13.7 | 14.5 | 15.0 | 13.9 | 14.0 |
| CollabUIAgents$_{mobile}$ | 7B/4A | 18.6 | 17.7 | 19.1 | 15.6 | 17.7 |
| CollabUIAgents$_{m \to web}$ | 7B/4A | **34.3** | **36.9** | **35.3** | **32.5** | **34.7** |

Table 9: Step Success Rates (SSR) of agent systems on different LLMs in the AutoWebBench environment.

of 1e-4 and training conducted over 3 epochs. During the preference optimization process, we adapt LoRA fine-tuning (Hu et al., 2022), and the model's context length is capped at 8,500, with a learning rate of 5e-6. We use bf16 precision, and the training is also carried out for 3 epochs.

In the multi-agent reinforcement learning process, the critic agent is played by GPT-4o-2024-08-06, with its temperature also set to 0.1, providing reward feedback to the rolled out actions. In the MARL phase, the rolling out temperatures of different UI agents are set to 0.1, 0.3, 0.5, and 0.8, respectively. This variation encourages diverse responses and stimulates different behaviors. We use iterative-DPO (Xiong et al., 2024) as the online reinforcement algorithm, and the multi-agent system is updated for each epoch. We train the multi-agent system within the AndroidWorld environment, where the environment seed is randomly generated. Continue MARL on Mind2Web uses instruction datasets autonomously collected with the pipeline in Appendix C. During testing, we maintain consistency with previous work (Rawles et al., 2025) by setting the seed to 30. All our experiments are conducted on 8 A100 80GB GPUs. The model's context length is capped at 8,500, with a learning rate of 5e-6. We use bf16 precision, and the training is carried out for 3 epochs where the off-policy rolling out is conducted at the beginning of each epoch.

### D.4 Data Collection for Agentic Fine-Tuning

We collect data from the AndroidWorld environment for agentic fine-tuning the base model into the base UIAgent. In this section, we provide details of the data collection process, including the questions list, the quantity of the collected data, and the prompts for data collection.

### D.4.1 Questions

The questions used for UI basic environmental knowledge generation are shown in Table 10.

| Type | Question |
|---|---|
| UI Understanding | 1. What is the purpose of the current UI? 2. What does the current UI aim to achieve? 3. Summarize the current interface in one paragraph. |
| Element Recognition | 1. What is the function of UI element X? 2. What information does UI element X provide? 3. What happens when click the UI element X? 4. What action is associated with UI element X? |

Table 10: Questions for UI basic environmental knowledge generation.

### D.4.2 Statistics of the Collected Data

The quantity of the collected data is shown in Table 11.

| Data Type | Number |
|---|---|
| Basic Environmental Data | 88,513 |
| Simple Instruction Data | 18,041 |
| Process Preference Data | 3,440 |

Table 11: Quantity of the collected data.

### D.4.3 Prompts

Prompts for data collection process in the agentic fine-tuning are shown in Figure 5, 6, 7, and 8.

### D.5 Prompts for Different Agents in the CR

Prompts for different agents in the CR are shown in Figure 8, 9 and 10.

### D.6 Prompts for CollabUIAgents Framework

Prompts for the multi-agent system trained in CollabUIAgents on mobile operation environments and web browsing environments are shown in Figure 11 and Figure 12, respectively, within the ReAct (Yao et al., 2023) style.

You are an agent who can operate an Android phone on behalf of a user.

Here is a list of descriptions for some UI elements on the current screen:

{ui_elements_description}

Please answer the following questions for all the UI elements above.

Questions = ('
'What is the purpose of the current UI?'
'Summarize the current interface in one paragraph.'
'What does the current UI aim to achieve?
)

Please format your response as follows:
'{{"Question": "What is the purpose of the current UI?", "Answer":"........"}}'
'{{"Question": "Summarize the current interface in one paragraph.", "Answer":"........"}}'
'{{"Question": "What does the current UI aim to achieve?", "Answer":"........"}}'

Your response:

Figure 5: The UI understanding prompt template.

You are an agent who can operate an Android phone on behalf of a user.

Here is a list of descriptions for some UI elements on the current screen:

{ui_elements_description}

Please answer the following questions for all the UI elements above.

Questions = (
'What is the function of UI element X ?'
'What information does UI element X provide ?'
'What happens when click the UI element ?'
'What action is associated with UI element X ?'
)

Please format your response as follows:
'{{"Question": "What is the function of UI element X?", "Answer":"........"}}'
'{{"Question": "What information does UI element X provide?", "Answer":"........"}}'
'{{"Question": "What happens when click the UI element X?", "Answer":"........"}}'
'{{"Question": "What action is associated with UI element X?", "Answer":"........"}}'

Your response:

Figure 6: The element recognition prompt template.

You are an agent who can operate an Android phone on behalf of a user.

Here is a list of descriptions for some UI elements on the current screen:

{ui_elements_description}

The action space of the agent: {action_space}

General guidance: {general_guidance}

Please propose diverse simple instructions (one-step tasks) as many as possible based on the agent\'s action space and the current UI elements above in the following format: (contains at least one but no more than two \'complete\' actions and no more than one \'answer\' action)'
'{{"Instruction": "......", "Response": "Reason: ... Action: {{"action_type":...}}"}}'

For example:
'{{"Instruction": "I need to start recording audio", "Response": "Reason: The recording settings are all configured, I need to click \'Apply\' to apply the current settings and start recording. Action: {{"action_type": "click", "index": 3}}"}}'
'{{"Instruction": "I want to select the M4a format for recording.", "Response": "Reason: The recording format has been set correctly. Action: {{"action_type": "status", "goal_status": "complete"}}"}}'

'Your response:

Figure 7: The instruction knowledge prompt template.

The current user goal/request is: {goal}

Here is a history of what you have done so far: {history}

Here is a list of descriptions for some UI elements on the current screen:

{ui_elements_description}

General guidance: {general_guidance}

Now you need to role-play a very clumsy agent that can only output incorrect answer (if you have no choice, you can make up a wrong action and reason) from the above list in the correct JSON format, following the reason why you do that.

Your answer should look like:
'Reason: ...Action: {{"action_type":...}}'

Your answer:

Figure 8: The prompt template for the adversarial agent.

You are a super-intelligent agent who can expertly operate an Android phone on behalf of a user.

Now, you need to act as a critic, evaluating the actions taken by other Android agents.

These agents receive user tasks and current Android interface information and then take the next step.

Your evaluation should be between [0,1]. A score close to 0 means the agent's action is useless or incorrect in achieving the user's task, a score close to 0.5 means you are uncertain whether the agent's decision is useful for achieving the user's task, and a score close to 1 means the agent's action is useful or correct in achieving the user's task.

The current user goal/request is: {goal}

Here is a history of what have done so far: {history}

Here is a list of descriptions for some UI elements on the current screen:

{ui_elements_description}

General guidance: {general_guidance}

Here are the next actions different agents would like to take: {agents_actions}

Please output each agent's score in the correct JSON format, following the reason why you think the agents' actions and reasons are correct or not, and ensuring that their actions are necessary and not redundant for achieving the user's goals when you give your scores.

Figure 9: The prompt template for the critic agent.

Your answer should look like:

'Reason: The goal is {{user goal}}...Score: {{"agent_id": score...}}'

Here are some demonstrations of evaluations:\n'

1. Reason: The goal is ... Agent 0 and Agent 2 attempt to scroll down to find additional options. This is a logical step given that no explicit save button is visible and the app might have additional options accessible through scrolling. Agent 1 decides to click the "Settings" button in hopes that it might lead to a menu with a save option. However, this seems less directly connected to saving the recording as the Settings menu is generally for configuration rather than saving recordings. Score: {{"agent_0": 0.9, "agent_1": 0.1, "agent_2": 0.9}}.

2. Reason: The goal is ... All agents (Agent 0, 1, and 2) have chosen to input the desired name "xxx.m4a" into the text field, However, user did not specify a name. This is the incorrect next step, ...Score: {{"agent_0": 0.2, "agent_1": 0.2, "agent_2": 0.2}}.

3. Reason: The goal is ... Historical information shows that the agent has taken the same action multiple times. I am unsure if taking the same action again is reasonable. Score: {{"agent_0": 0.5, "agent_1": 0.5, "agent_2": 0.5}}.

Now output each agent's score.

Your answer must in the format:

'Reason: The goal is {{user goal}}...Score: {{"agent_id": score, "agent_id": score, "agent_id": score}}'

Your Evaluation:

Figure 10: The few-shot prompt template for the critic agent.

The current user goal/request is: {goal}

Here is a history of what you have done so far: {history}

Here is a list of descriptions for some UI elements on the current screen:

{ui_elements_description}

General Guidance: {general_guidance}

Now output an action from the above list in the correct JSON format following the reason

why you do that. Your answer should look like:

'Reason: ...Action: {{"action_type":...}}'

Your answer:

Figure 11: The prompt template for mobile operation.

<html > {html_content} </html >

You are a helpful assistant that can assist with web navigation tasks. You are given a

simplified html webpage and a task description. Your goal is to complete the task. You can

use the provided functions below to interact with the current webpage.

#Provided functions: {action_space}

#Previous commands: {previous_commands}

#Window tabs: {exist_window_tabs_with_pointer_to_current_tab}

#Current viewport (pages): {current_position} / {max_size}

#Task: {task_description}

You should output one command to interact to the currrent webpage. You should add a brief

comment to your command to explain your reasoning and thinking process.

Figure 12: The prompt template for web browsing.

