# OpenReview forum: "Advancing Language Multi-Agent Learning with Credit Re-Assignment for Interactive Environment Generalization"
_colmweb.org/COLM/2025/Conference — COLM 2025_

### Official Review · Reviewer_Mpaq · 2025-04-23

**Rating:** 6
**Confidence:** 4
**Ethics Flag:** 1

**Summary:**

The paper presents CollabUIAgents, a multi-agent reinforcement learning algorithm which aims to better train agents in collaborative environments. They achieve this through 3 main techniques, mainly credit reassignment, which uses an LLM to provide fine-grained rewards for steps in collaboration rather than a simple 1 or 0 final reward. In addition, they use preference optimization with adversarial outputs, and provide the "edge update" trick.

**Questions To Authors:**

- Do you think that a policy trained on critic written rewards can exceed the capabilities of the critic? If so, can you provide experimental evidence or use a wider array of critics?
- How does this compare to directly distilling gpt-4o?
- How do you know if a prompt is good for the llm critic? Do you experiment with different critic prompts?
- Can you add one or two more training runs which vary the base model to show that other base policies see similar gain or how they end up landing vs. Gpt4? Especially larger models?
- Do you have a sense of noise in performance? I'm especially curious about table 3, where I feel some differences may be within noise range.
- Could you explain the "edge update" trick in more detail and with more ablations? I don't fully understand what it entails from what is written or figure 2. The paper mentions that edges are "randomly" updated - how much does this matter, is probabilily < 1  optimal or is it for efficiency?
- Can you provide sample outputs from the learned policies and the corresponding critic outputs? I want to see how the agents evolve - what they learn, and especially if they differ from gpt-4.

**Reasons To Accept:**

This paper pushes in an extremely important direction and shows a solid proof of concept for many good ideas. They show very strong performance with a small model. Their training setup seems solid, and overall I do believe in the core concept being presented.

**Reasons To Reject:**

Reaching gpt-4 performance is impressive, but honestly a bit suspicious considering that 4o is in fact also the critic model. I'm not sure if these are slightly different GPT versions, but it suggests to me that this may essentially be distilling the critic into the agents. This isn't necessarily a bad thing, but it significantly changes the framing of the method and raises other questions (can we achieve the same performance through directly training on gpt-4 outputs?). I'm also concerned that this detail isn't mentioned until appendix section c.3.2. This should definitely be mentioned in the main text.

I'm also not convinced by the novelty in the method. Training with AI feedback is well established, so the main novelty is applying this to multi-agent environments. However, there are not enough ablations for this - what happens as we vary the critic? Can we use the model to critique itself? Can we get better results from pure distillation if we have access to a critic like gpt-4o?

---

> ### Author Response · Authors · 2025-06-01
> **(1/n) Responses by Authors**
>
> Thanks for your careful and insightful reviews.
>
> > `Weakness 1`: Reaching GPT-4 performance is impressive, but this may essentially be distilling the critic into the agents. Can we achieve the same performance by directly training on GPT-4 outputs?
>
> Thank you for your insightful comments. In our proposed CollabUIAgents framework, GPT-4o is used as the Critic Agent. However, its role is not to directly provide action outputs for the agents to imitate. Instead, the Critic Agent's core function is to provide fine-grained process rewards for our multi-agent credit re-assignment (CR) strategy, based on its understanding of the environment state, interaction history, and agent behaviors. **Our framework is a reinforcement learning paradigm where agents learn by interacting with the environment and receiving these LLM-guided rewards and preferences, rather than simply replicating GPT-4o's outputs**.
>
> Furthermore, **we introduce 'Distilling' baselines**, which were train via supervised fine-tuning (SFT) directly on data where GPT-4 provided the correct answers (i.e., successful trajectories/actions). We ensure that the volume of data used for this SFT-based distillation was consistent with the amount of preference data used for DPO. The results from the expanded Table 1 are as follows:
>
> | Method | Params  | Input  |   SR(Androidworld)  | SR(MMiniWoB++)  | ∆ Generalization  |
> |-------------|----------|--------------|----------| ----------| ----------|
> | **Closed-Scoure** |
> | M3A (GPT4) | N/A | Text | **30.6** | 59.7 | - |
> | M3A (GPT4) | N/A | Multimodal | 25.4 | **67.7** | - |
> | **Open-Scoure** |
> | Qwen2 | 7B | Text |  6.2 | 12.9  | - |
> | Qwen2 VL | 2B | Multimodal | 0.0 | 10.0 | - |
> | Distilling (Qwen2) | 7B | Text | 12.3 | 35.8 | 22.9 |
> | Distilling (Qwen2) | 4x7B | Text | 15.2 | 38.0 | 25.1 |
> | UIAgent (Qwen2) | 7B | Text | 18.9  | 48.4  | 35.5 |
> | UIAgent (Qwen2) | 4x7B | Text | 21.4  | 53.2  | 40.3 |
> | CollabUIAgents (Qwen2) | 4x7B | Text | **29.3**  | **61.2**  | **48.3** |
>
> The Distilling (Qwen2) 7B model, trained via this SFT approach on GPT-4's correct outputs, achieved SRs of only 13.3% and 35.8%. Even the Distilling (Qwen2) 4x7B model only reached SRs of 15.2% and 38.0%. **These results strongly indicate that the performance achieved by our CollabUIAgents framework, through its novel MARL with CR strategy**, significantly surpasses methods based on direct SFT distillation from a strong teacher model like GPT-4
>
> > `Weakness 2`: The critical detail that the critic model is GPT-4o, it was not mentioned until Appendix C.3.2.
>
> Thank you for your feedback. To ensure this crucial information is more accessible and prominently featured, we will revise the manuscript to include the identity of the critic model much earlier. Specifically, **we will add this detail in Section 3.2** 'Language Multi-Agent RL,' within the 'Credit Re-Assignment' subsection, where the critic agent and its role in assigning process rewards are first introduced.
>
>
> > `Weakness 3`: There are not enough ablations (vary the critic).
>
> To address your questions about varying the critic and the possibility of self-critique, **we conduct a new ablation study**. The results are as follows:
>
> | Method  |   SR(Androidworld)  | SR(MMiniWoB++)  | Critic  |
> |-------------|----------|--------------|----------|
> |Single-Agent Systems|
> | Qwen2  |  6.2 | 12.9  | - |
> | UIAgent (Qwen2) | 18.9  | 48.4  | GPT-4 |
> | UIAgent (Qwen2) | 10.7  |  19.5 | Qwen2 (self-critic) |
> |Multi-Agent Systems (n = 4)|
> | Qwen2  |  8.6 | 16.1  | - |
> | UIAgent (Qwen2) | 21.4  | 53.2  | GPT-4 |
> | UIAgent (Qwen2) | 12.5 |  26.1 | Qwen2 (self-critic) |
>
> **The capability of the critic model significantly impacts agent performance**. Using GPT-4 as the critic for preference data generation yielded substantially higher success rates (e.g., 18.9% on AndroidWorld for single-agent, 21.4% for multi-agent ensemble) compared to using Qwen2 for self-critique (10.7% and 12.5%, respectively). This performance disparity underscores why self-critique with a less capable model like Qwen2 is insufficient, it lacks the nuanced understanding and accurate feedback generation capabilities of a stronger critic like GPT-4. **A less capable self-critic cannot effectively identify subtle errors or provide high-quality preference signals, leading to suboptimal agent policies**. However, it can still provide valuable rewards for the agent, and it's possible for the agent to surpass the critic on some tasks.

---

> > ### Comment · Reviewer_Mpaq · 2025-06-06
> >
> > Thanks for the response and the additional ablations!
> >
> > Yes, I understand that using GPT-4o as a critic is different to direct knowledge distillation. What I mean is this: **is CollabUIAgents a better way of performing distillation than SFT?** We can view SFT as one option of trying to distill knowledge from a teacher, however, this does not mean that it is the best way to distill. CollabUIAgents may also be considered a way of distilling knowledge from a teacher, though it may be better and more principled.
> >
> > A common question in the field is whether asking GPT-4o to act as a critic would allow the student to ever surpass the teacher - the belief is that critiquing an answer is easier than generating one. The common consensus seems to be yes, but it's a good litmus test for methods like this.
> >
> > I'm glad you provided the ablation with self-critique. The important thing to me is not whether the self-critique surpasses 4o critique, but whether it improves at all, which it does!

---

> > > ### Author Response · Authors · 2025-06-06
> > >
> > > **Thank you again for your thoughtful follow-up comments!**
> > >
> > > Our framework leverages the critic's capabilities to provide high-quality preference signals and process-level rewards, which guide the agents' learning in a reinforcement learning paradigm. This allows for a more nuanced and adaptive learning process than simply replicating successful outputs via SFT. **The significant performance gap between our CollabUIAgents and the "Distilling" baselines (as shown in the expanded Table 1 in our previous response) strongly supports this argument, demonstrating that our approach is superior to direct SFT-based distillation**.
> > >
> > > We are also pleased that our self-critique ablation was valuable. We agree that the improvement, even if not surpassing the GPT-4o critique, is a significant indicator of the method's robustness and its ability to learn valuable signals even from a less capable self-critic. This suggests potential for future work on iterative self-improvement.
> > >
> > > Furthermore, **we would like to highlight that the "w/ PO → RFT" row in our ablation study** (Table 3, repeated below for convenience) serves as another form of supervised fine-tuning using data generated through our framework. "PO → RFT" (Preference Optimization → Reject Fine-Tuning) involves performing rejective fine-tuning based on the CR rewards, effectively filtering unrewarded data. This process is essentially performing SFT on the preferred data, which are the successful trajectories identified and rewarded by the critic. The success rates for "w/ PO → RFT" are 23.2% for AndroidWorld and 54.8% for MMiniWoB++. These results, when compared to the full CollabUIAgents model (29.3% and 61.2%), **demonstrate that while SFT on curated data can provide a strong baseline, the full CollabUIAgents framework with its continuous reinforcement learning and credit re-assignment mechanism achieves even higher performance, reinforcing our claim of its superiority over pure SFT-based approaches**.
> > >
> > > Here is the ablation study table for reference:
> > > | Method |   SR(Androidworld)  | SR(MMiniWoB++)  |
> > > |-------------|----------|--------------|
> > > | **Single-Agent Systems** |
> > > | Qwen2 | 6.2 | 12.9 |
> > > | UIAgent (Qwen2) | 18.9 | 48.4 |
> > > | **Multi-Agent Systems** |
> > > | Qwen2 | 8.6 | 16.1 |
> > > | UIAgent (Qwen2) | 21.4 | 53.2 |
> > > | CollabUIAgents_mobile | 29.3 | 61.2 |
> > > | **w/ PO → RFT** | 23.2 | 54.8 |
> > > | w/o CR | 25.0 | 56.4 |
> > > | w/o Edge Update | 27.6 | 58.1 |
> > > | CollabUIAgents_m→web | 26.7 | 58.1 |
> > >
> > > We appreciate your continued engagement and valuable feedback, which has helped us to clarify and strengthen our paper. We are fully prepared to address any remaining questions you may have and welcome further discussion!

---

> ### Author Response · Authors · 2025-06-01
> **(2/n) Responses by Authors**
>
> > `Question 1`: Can you provide experimental evidence or use a wider array of critics?
>
> Our agents learn generalizable policies by exploring and optimizing cumulative process rewards within a multi-agent reinforcement learning (MARL) framework. Experimentally, our CollabUIAgents (built on Qwen2 7B and guided by a GPT-4o critic) demonstrates performance comparable to or even exceeding that of GPT-4 acting as an agent. **As detailed in our responses to Weakness 1 and 3**, CollabUIAgents also significantly outperforms direct distillation from GPT-4 outputs and benefits from a strong critic (as opposed to a self-critic).
>
> > `Question 2`: How does this compare to directly distilling gpt-4o?
>
> Thank you for your question regarding how our method compares to directly distilling GPT-4o. We have addressed this specific comparison in **detail in our response to Weakness 1**.
>
> > `Question 3`: Do you experiment with different critic prompts?
>
> Thank you for your questions regarding the LLM critic's prompt. Our initial approach to designing a "good" prompt for the LLM critic focused on several key principles: (1) **clarity and specificity of role**, the prompt clearly defines the critic's task; (2) **comprehensive context provision**, the prompt ensures the critic receives all necessary information, including the task, historical actions, current UI element descriptions, and the specific actions proposed by the agents being evaluated; (3) **structured output format**, we guide the critic to produce outputs in a consistent JSON format, including scores and reasoning, which is crucial for systematic reward assignment and preference data synthesis. Furthermore, we posit that powerful LLMs like GPT-4o, due to their extensive pre-training and rich world knowledge, exhibit a degree of robustness to minor variations in prompt phrasing. They can often infer the underlying intent well, especially when the core task and context are clearly provided.
>
> We did experiment with different critic prompts to assess their impact. The results are presented below:
>
> | Method | Params  | Input  |   SR(Androidworld)  | SR(MMiniWoB++)  | Critic prompt  |
> |-------------|----------|--------------|----------| ----------| ----------|
> | UIAgent (Qwen2) | 7B | Text | 18.9  | 48.4  | Original |
> | UIAgent (Qwen2) | 7B | Text | 17.8  | 46.3  | Modified |
>
> **The modified prompt was created by altering the original to be more concise, primarily providing only the required output format and essential contextual information (such as the current user goal, interaction history, and UI state)**. The UIAgent (Qwen2) with the Modified critic prompt achieved SRs of 17.8% on AndroidWorld and 46.3% on MMiniWoB++. This performance is only slightly lower than the results obtained with our Original prompt. **This relatively minor difference in end-task performance suggests that the strong critic model is indeed quite robust** to reasonable variations in the prompt structure. Furthermore, to quantify the consistency of the critic's evaluations, we analyzed the rewards assigned by the critic under these different prompts. Our statistics show that approximately 94.2% of the rewards assigned were consistent across the original and modified prompts for the same input scenarios.
>
>
> > `Question 4`: Can you add one or two more training runs which vary the base model to show that other base policies see similar gain or how they end up landing vs. Gpt4? Especially larger models?
>
> Thank you for your suggestion. To address your request, **we conducted additional experiments using LLaMA2 7B** as an alternative base model. The results are presented below:
>
> | Method | Params  | Input  |   SR(Androidworld)  | SR(MMiniWoB++)  | ∆ Generalization  |
> |-------------|----------|--------------|----------| ----------| ----------|
> | **Open-Scoure** |
> | Qwen2 | 7B | Text |  6.2 | 12.9  | - |
> | LLaMA2 | 7B | Text | 3.5 | 11.2 |  |
> | Qwen2 VL | 2B | Multimodal | 0.0 | 10.0 | - |
> | Qwen2.5 VL | 3B | Multimodal | 10.1 | 18.7 | - |
> | UIAgent (Qwen2) | 7B | Text | 18.9  | 48.4  | 35.5 |
> | UIAgent (LLaMA2) | 7B | Text | 15.1 | 43.7 | 32.7 |
>
> The raw LLaMA2 7B model started with lower performance, After applying our agentic fine-tuning, UIAgent (LLaMA2) improved to SR 15.1% (AndroidWorld) and 43.7% (MMiniWoB++), showing gains of +11.6 and +32.5 respectively. These results indicate that our framework brings substantial and comparable performance gains to different base models. **We acknowledge your interest in seeing results with larger base models. Ideally, we would explore a wider range of base models, including larger ones. However, our experiments, especially those involving the full MARL pipeline of CollabUIAgents, are very intensive in terms of time and computational resources**. Due to these significant constraints, we had to limit the scope of base model variations for the full framework evaluation.

---

> > ### Comment · Reviewer_Mpaq · 2025-06-06
> >
> > Thanks for the additional experiments and quick response!

---

> > > ### Author Response · Authors · 2025-06-06
> > >
> > > We are very thankful for your detailed review and happy to provide any further clarifications or details you may require !

---

> ### Author Response · Authors · 2025-06-01
> **(3/n) Responses by Authors**
>
> > `Question 5`: Do you have a sense of noise in performance? I'm especially curious about table 3, where I feel some differences may be within noise range.
>
> For all our experiments, including the ablation studies in Table 3, the reported results are based on evaluations **conducted with a fixed seed**, as detailed in Appendix C.3.2. This approach ensures fair and reproducible comparisons between different models and configurations, consistent with practices in related work. Each full MARL training run for our LLM-based multi-agent system is computationally intensive, which limited our ability to perform numerous identical training runs for each reported figure to establish statistical variance.
>
> > `Question 6`: Could you explain the "edge update" trick in more detail and with more ablations? I don't fully understand what it entails from what is written or figure 2. The paper mentions that edges are "randomly" updated - how much does this matter, is probabilily < 1 optimal or is it for efficiency?
>
> **The 'Edge Updates' are random in the sense that for each training iteration (or batch of trajectories), a new DAG communication subgraph is sampled from the set of potential DAGs for the ∣G∣ agents.** As stated on page 5 (lines 209-212), it alleviates the significant overhead of learning an optimal, static communication graph from an enormous combinatorial space. More importantly, by exposing agents to diverse communication structures during training, we "encourage agents to learn the awareness of multi-agent collaboration and adapt to diverse message networks rather than being rigid in locally optimal DAG pattern". This promotes robustness and generalizability in their collaborative policies.
>
> In our Ablation Study (Section 4.3, Table 3). The results clearly demonstrate that 'CollabUIAgentsmobile' (which includes edge updates) achieves higher success rates (SR 29.3 in AndroidWorld, 61.2 in MMiniWoB++) compared to the 'w/o Edge Update' variant (SR 27.6 in AndroidWorld, 58.1 in MMiniWoB++). To address your concern about clarity, **we will revise Section 3.2 to more explicitly connect the 'random' sampling** to these conceptual and empirical justifications, ensuring its role and benefits are better highlighted.
>
>
> > `Question 7`: Can you provide sample outputs from the learned policies and the corresponding critic outputs? I want to see how the agents evolve - what they learn, and especially if they differ from gpt-4.
>
> Our manuscript already includes **an initial example in Appendix A** (Figure 4, page 17), which illustrates task execution steps in the AndroidWorld environment. As stated on lines 694-695, this example "demonstrates that the reward from the CR process is correctly identified for each action." This figure shows agent actions and the associated rewards assigned by our GPT-4o critic through the Credit Re-Assignment (CR) process.
>
> Through our MARL framework with CR, agents learn to generate sequences of actions that are preferred by the GPT-4o critic for effective task completion and collaboration. They adapt to dynamic communication structures (due to Edge Updates) and learn generalizable strategies. Crucially, our results indicate that the learned policies can indeed differ from or even surpass the direct capabilities of a GPT-4 baseline, especially in generalization. As noted on lines 79-80, our system "achieves performance comparable to the guidance LLM used in the CR, GPT-4... in training environments, and even better in an unseen environment." This suggests that by learning from preferences within an interactive MARL setup, our agents can develop policies that are highly effective and sometimes more robustly generalizable than applying a monolithic GPT-4 model directly to the task. The agents learn to align with the critic's preferences over explored actions, rather than directly mimicking a specific set of GPT-4 demonstrated trajectories.

---

> ### Comment · Reviewer_Mpaq · 2025-06-06
>
> Thanks for the detailed responses and additional experiments! I will adjust my score to a 6.

---

> ### Author Response · Authors · 2025-06-06
>
> Thank you very much for your kind words regarding our responses.
>
> We are truly grateful for your thorough review and constructive feedback throughout this process. **Your insightful comments have been invaluable in helping us to significantly improve the quality and clarity of our manuscript!**
>
> We remain committed to addressing all feedback to ensure the final version of our paper is as strong as possible and **have specifically conducted additional experiments exploring different numbers of agents to demonstrate the robustness and generalizability of our method**. Our updated experimental results table (extended Table 1) is shown below:
>
> | Method | Params  | Input  |   SR(Androidworld)  | SR(MMiniWoB++)  | ∆ Generalization  |
> |-------------|----------|--------------|----------| ----------| ----------|
> | **Open-Scoure** |
> | Qwen2 | 7B | Text |  6.2 | 12.9  | - |
> | Qwen2 | 4x7B | Text |  4.3 | 15.2  | - |
> | Qwen2 VL | 2B | Multimodal | 0.0 | 10.0 | - |
> | Qwen2 VL | 4x2B | Multimodal | 0.0 | 11.5 | - |
> | Qwen2.5 VL | 3B | Multimodal | 10.1 | 18.7 | - |
> | Qwen2.5 VL | 4x3B | Multimodal | 12.6 | 21.3 | - |
> | DigiRL (Qwen2.5 VL) | 3B | Multimodal | 22.3 | 35.2 | 16.5 |
> | DigiRL (Qwen2.5 VL) | 4x3B | Multimodal | 20.1  | 38.7  | 20.0 |
> | UIAgent (Qwen2) | 7B | Text | 18.9  | 48.4  | 35.5 |
> | UIAgent (Qwen2) | 4x7B | Text | 21.4  | 53.2  | 40.3 |
> | UIAgent (Qwen2) | 6x7B | Text | 18.9  | 54.3  | 41.5 |
> | CollabUIAgents (Qwen2) | 4x7B | Text | 29.3  | **61.2**  | **48.3** |
> | CollabUIAgents (Qwen2) | 6x7B | Text | **32.7**  | 59.7  | 46.9 |
>
> These results highlight that the performance advantage of CollabUIAgents stem from its unique MARL framework that effectively learns and fosters complex collaboration among agents.
>
> Hope for your continued valuable insights and favorable consideration!

---

> ### Author Response · Authors · 2025-06-09
>
> Dear Reviewer Mpaq,
>
> We are grateful for your constructive comments. We have revised the manuscript accordingly. Please let us know if you have any additional questions.
>
> Thank you again for your time and expertise.
>
> Sincerely,
>
> The Authors

---

### Official Review · Reviewer_sVoJ · 2025-04-29

**Rating:** 4
**Confidence:** 4
**Ethics Flag:** 1

**Summary:**

This paper employs multi-language-agents systems to solve problems in mobile and web environments. To mitigate the sparse reward issue, the paper designs an LLM-based credit re-assignment technique. Experiments show that this framework achieves comparable results with advanced closed-source large models like GPT-4 based on the small model Qwen2.

**Reasons To Accept:**

1. The application of MARL to mobile/web automation is relatively under-explored and presents practical value.
2. The proposed credit re-assignment technique could potentially benefit other cooperative multi-agent scenarios if properly validated.

**Reasons To Reject:**

1. **Questionable Multi-Agent Contribution**:
    - The "multi-agent" framework appears functionally equivalent to a single-agent system with role-specific prompts and voting. The agents lack true coordination.
    - The choice of agent roles/numbers relies heavily on human priors, with no justification or ablation studies.
2. **Methodological Issues**:
    - The so-called "adversary" does not engage in meaningful opposition; it merely generates inferior actions as negative samples. This may confuse with robust RL and MARL literature [1,2,3,4].
    - The core learning part (Eq (8)) is not clearly explained, with multiple symbols undefined (Although I found some of them in Appendix B.1).
    - Edge updates are described as "random" without theoretical or empirical justification, raising doubts about its role.
3. **Clarity and Rigor**:
    - Key components (e.g., Eq. 8, "conversation rounds") are ambiguously defined.
    - Section 3.3 is superficial, stating obvious facts about fine-tuning without linking to the proposed method.
    - Typographical errors (e.g., line 319, faces → face) undermine professionalism.
4. **Lack of Validation**:
    - The method’s generality is claimed but untested. No experiments on simpler tasks (e.g., grid-worlds) validate its core mechanisms before complex mobile/web deployments.

References

[1] Pinto et al. Robust adversarial reinforcement learning. 2018.

[2] Bukharin et al. Robust Multi-Agent Reinforcement Learning via Adversarial Regularization: Theoretical Foundation and Stable Algorithms. 2023.

[3] Yuan et al. Robust Multi-agent Coordination via Evolutionary Generation of Auxiliary Adversarial Attackers. 2023.

[4] Lee et al. Wolfpack Adversarial Attack for Robust Multi-Agent Reinforcement Learning. 2025.

---

> ### Author Response · Authors · 2025-06-01
> **(1/n) Responses by Authors**
>
> Thanks for your careful and insightful reviews.
>
> > `Weakness 1`: The "multi-agent" framework ‘ppears functionally equivalent to a single-agent system with role-specific prompts and voting. The agents lack true coordination.
>
> Thank you very much for your insightful comments. We understand that from a functional performance perspective, a highly optimized single-agent system (via role-specific prompts and voting) might exhibit similarities to a multi-agent system in some aspects. However, we wish to emphasize that CollabUIAgents fundamentally differs from such single-agent setups in its learning mechanisms, dynamic interactions, the emergent nature of coordination, and its pursuit of generalizability. **There are several differences**:
> - **Our core is MARL-driven Policy Learning, Not Fixed Role Prompts**: CollabUIAgents is a Multi-Agent Reinforcement Learning (MARL) framework. Each agent's policy is not based on fixed role prompts but is learned and optimized through interaction with the environment and guided by our proposed Credit Re-Assignment (CR) strategy (as shown in Equation 8 of the paper). Agents are "role-free," and their objective is to learn collaborative behaviors that maximize common rewards, rather than executing pre-scripted roles. This capacity for learning and adaptation is difficult to replicate with single-agent fixed-prompt methods.
> - **Dynamic Communication Structure and Adaptability**: We introduced the "Edge Updates" technique, meaning the communication network among agents is not static but is randomly updated. This compels agents to "learn to adapt to diverse message networks rather than being rigid in locally optimal DAG patterns". Our ablation study (Table 3) shows that removing edge updates ("w/o Edge Update") impairs performance, demonstrating the importance of agents learning to adapt to dynamic communication structures, which significantly differs from a single-agent model using fixed "roles."
>
> **Regarding 'true coordination’**, this is primarily fostered by:
> - **Our novel Credit Re-Assignment (CR) strategy**: This uses an LLM critic to provide fine-grained process rewards for intermediate collaborative actions at both agent and conversation round levels. This rewards the process of collaboration, guiding agents to learn how to coordinate effectively, beyond just final outcomes.
> - **Preference-based policy learning**: Agents are optimized using preference data synthesized from these CR rewards, directly encouraging policies that generate rewarded (i.e., more collaborative) actions.
> - **Structured interaction**: Agents utilize shared context and messages within a DAG communication network, allowing sequential contribution before a final action is aggregated via voting. Thus, the inputs to the vote are already shaped by learned, coordinated policies.
> Empirically, our ablation studies (Table 3) demonstrate that these MARL and CR mechanisms are crucial. CollabUIAgents significantly outperforms a 'Base UIAgent (n=4)' (multi-agent without our advanced MARL/CR) and a version 'w/o CR', underscoring that our specific coordination-focused designs contribute substantially beyond merely using multiple agents or simpler learning approaches.
>
> We will ensure these distinctions are more explicitly highlighted in the revised manuscript. Thank you again for your valuable feedback.
>
>
> > `Weakness 2`: The choice of agent roles/numbers relies heavily on human priors, with no justification or ablation studies.
>
> Thank you for your feedback regarding agent roles and numbers. **A core design principle of CollabUIAgents is that our agents are role-free**. We intend for any specialized behaviors or 'roles' to emerge organically through the Multi-Agent Reinforcement Learning (MARL) process, driven by our Credit Re-Assignment (CR) strategy  and preference learning. **Our framework also features 'Edge Updates’**, allowing the system to dynamically adapt its internal communication structure for a given set of agents, fostering adaptability rather than relying on fixed configurations. Regarding the number of agents (n=4) and conversation rounds (m=3), this configuration was selected to **balance collaborative potential and communication depth against significant computational costs and context length limitations** associated with LLM-based MARL. Ideally, **the number of agents could even be explored more dynamically by the system itself**. However, due to current computational resource constraints, we manually set this for our main experiments.

---

> ### Author Response · Authors · 2025-06-01
> **(2/n) Responses by Authors**
>
> > `Weakness 3`: The so-called "adversary" does not engage in meaningful opposition; it merely generates inferior actions as negative samples. This may confuse with robust RL and MARL literature [1,2,3,4].
>
> Thank you for your insightful comment regarding the term 'adversarial agent' and its potential for confusion with the robust RL and MARL literature. **You are correct that the role of our 'adversarial agent' differs from adversaries in traditional robust reinforcement learning**. In CollabUIAgents, π_adv is specifically designed to generate negative or suboptimal action samples. These are paired with preferred actions​ identified via our Credit Re-Assignment mechanism) to create preference data. Its purpose is to provide a clear contrast for learning, rather than to engage in 'meaningful opposition' for robust training in a game-theoretic sense. We acknowledge that this usage could be confusing. **To address this, we will add clarifying sentences, explicitly distinguishing its function** in our framework from that of adversaries in the robust RL literature you cited, and explaining our rationale for using the term.
>
>
> > `Weakness 4`: The core learning part (Eq (8)) is not clearly explained, with multiple symbols undefined (Although I found some of them in Appendix B.1).
>
> Thank you for your valuable feedback on the clarity of Equation (8) and its associated symbols. Equation (8) formulates the core learning objective for each agent in our MARL framework, adapting the Direct Preference Optimization (DPO) loss. To enhance clarity in the revised manuscript, **we will expand the explanation accompanying Equation (8) in Section 3.2 to more explicitly describe its components**.
>
> > `Weakness 5`: Edge updates are described as "random" without theoretical or empirical justification, raising doubts about its role.
>
> Thank you for your question regarding the 'Edge Updates' mechanism. **The 'Edge Updates' are indeed random in the sense that for each training iteration (or batch of trajectories), a new DAG communication subgraph is sampled from the set of potential DAGs for the ∣G∣ agents**. As stated on page 5 (lines 209-212), it alleviates the significant overhead of learning an optimal, static communication graph from an enormous combinatorial space. More importantly, by exposing agents to diverse communication structures during training, we "encourage agents to learn the awareness of multi-agent collaboration and adapt to diverse message networks rather than being rigid in locally optimal DAG pattern". This promotes robustness and generalizability in their collaborative policies.
>
> | Method |   SR(Androidworld)  | SR(MMiniWoB++)  |
> |-------------|----------|--------------|
> | **Single-Agent Systems** |
> | Qwen2 | 6.2 | 12.9 |
> | UIAgent (Qwen2) | 18.9 | 48.4 |
> | **Multi-Agent Systems** |
> | Qwen2 | 8.6 | 16.1 |
> | UIAgent (Qwen2) | 21.4 | 53.2 |
> | CollabUIAgents_mobile | 29.3 | 61.2 |
> | w/ PO → RFT | 23.2 | 54.8 |
> | w/o CR | 25.0 | 56.4 |
> | w/o Edge Update | 27.6 | 58.1 |
> | CollabUIAgents_m→web | 26.7 | 58.1 |
>
> In our Ablation Study, the results clearly demonstrate that 'CollabUIAgentsmobile' (which includes edge updates) achieves higher success rates (SR 29.3 in AndroidWorld, 61.2 in MMiniWoB++) compared to the 'w/o Edge Update' variant (SR 27.6 in AndroidWorld, 58.1 in MMiniWoB++). To address your concern about clarity, we will revise Section 3.2 to more explicitly connect the 'random' sampling to these conceptual and empirical justifications, ensuring its role and benefits are better highlighted.
>
>
> > `Weakness 6`: Key components (e.g., Eq. 8, "conversation rounds") are ambiguously defined.
>
> Thank you for your feedback on the clarity of key components like Equation (8) and 'conversation rounds.' As we will elaborate in the manuscript in response to your earlier comment, Equation (8) represents the Direct Preference Optimization (DPO) loss tailored for our MARL framework. Most symbols are defined in Section 2 or near Equation (8), with standard DPO components also detailed in Appendix B.1 and originating from the cited DPO literature. We will enhance the explanation of Equation (8) and ensure all its symbols are explicitly defined or clearly cross-referenced in the main text. Regarding 'Conversation Rounds' (m), the concept of 'conversation rounds' (m) is introduced in Section 2. A 'conversation round' refers to a cycle within a single environment time step t, where each of the n agents, operating sequentially as per the DAG communication structure, outputs an action proposal and messages. **We will carefully review and refine the definitions and explanations of both Equation (8) and 'conversation rounds' in the revised manuscript to ensure their clarity**.

---

> ### Author Response · Authors · 2025-06-01
> **(3/n) Responses by Authors**
>
> > `Weakness 7`: Section 3.3 is superficial, stating obvious facts about fine-tuning without linking to the proposed method.
>
> Section 3.3 outlines two primary approaches for adapting CollabUIAgents to new environments, both of which are directly tied to our method's capabilities and are empirically demonstrated:
> - **Direct Transfer**: We posit that agents trained with CollabUIAgents are suitable for direct transfer due to the generalizable collaborative behaviors fostered by our core mechanisms (e.g., the Credit Re-Assignment (CR) strategy, role-free learning, and edge updates). The multi-agent setup is also expected to contribute to robustness. This is empirically supported by our results for 'CollabUIAgentsmobile' (trained on mobile) when directly applied to web environments (Table 2: Mind2Web SSR 16.2, AutoWebBench SSR 17.7 ).
> - **Continual MARL**: This subsection explicitly states that agents "can undergo further training using the MARL with the CR strategy on the new environment". This refers directly to the re-application of our core CollabUIAgents learning paradigm (detailed in Section 3.2), not generic fine-tuning. The significantly improved performance of 'CollabUIAgentsm→web' in Table 2 (Mind2Web SSR 30.7, AutoWebBench SSR 34.7 ), which results from this continual MARL process, demonstrates the effectiveness of this specific adaptation strategy using our proposed method.
>
> To address your concerns and enhance clarity, we will revise Section 3.3 to more explicitly link the 'Direct Transfer' capability to the generalization-promoting features inherent in CollabUIAgents and add a forward reference from Section 3.3 to the experimental results in Section 4 that validate these adaptation strategies with CollabUIAgents.
>
>
> > `Weakness 8`: Typographical errors (e.g., line 319, faces → face) undermine professionalism.
>
> Thank you for meticulously pointing out the typographical errors in our manuscript. We have taken your feedback seriously and **will conduct a thorough proofreading of the entire manuscript to correct all typographical and grammatical errors**.
>
> > `Weakness 9`: The method’s generality is claimed but untested. No experiments on simpler tasks (e.g., grid-worlds) validate its core mechanisms before complex mobile/web deployments.
>
> We claim generality primarily in the context of complex, language-rich interactive environments. Our experiments demonstrate this by successfully applying and transferring CollabUIAgents across distinct and challenging domains: mobile operations (AndroidWorld, MobileMiniWoB++) and web Browse (Mind2Web, AutoWebBench). The framework's ability to generalize, **for example**, from mobile to web environments (as shown by 'CollabUIAgentsm→web' in Table 2 ), directly supports this claim within these complex UI-based tasks. Our core mechanisms, such as the Credit Re-Assignment (CR) strategy utilizing LLM knowledge  and role-free agent learning, are specifically designed to foster such generalizable behaviors in these types of environments. We aim to directly validate CollabUIAgents and its LLM-driven CR mechanism  in sophisticated, interactive tasks where LLMs' understanding of language, UI semantics, and general world knowledge can be most effectively leveraged for both agent control and credit assignment. These are environments where the challenges of sparse rewards and generalization are particularly acute for LLM-based agents, and where our method aims to make a significant contribution. Simpler environments often do not offer the same richness of interaction or semantic complexity where the benefits of an LLM-guided CR strategy would be most apparent. To improve clarity, **we will refine the discussion in our manuscript to more precisely define the scope of "generality" we address. Thank you for your comments**.

---

> ### Author Response · Authors · 2025-06-08
> **(4/n) Responses by Authors**
>
> We are truly grateful for your thorough review and constructive feedback throughout this process. **Your insightful comments have been invaluable in helping us to significantly improve the quality and clarity of our manuscript!**
>
> We remain committed to addressing all feedback to ensure the final version of our paper is as strong as possible and **have specifically conducted additional experiments exploring different numbers of agents to demonstrate the robustness and generalizability of our method**. Our updated experimental results table (extended Table 1) is shown below:
>
> | Method | Params  | Input  |   SR(Androidworld)  | SR(MMiniWoB++)  | ∆ Generalization  |
> |-------------|----------|--------------|----------| ----------| ----------|
> | **Open-Scoure** |
> | Qwen2 | 7B | Text |  6.2 | 12.9  | - |
> | Qwen2 | 4x7B | Text |  4.3 | 15.2  | - |
> | Qwen2 VL | 2B | Multimodal | 0.0 | 10.0 | - |
> | Qwen2 VL | 4x2B | Multimodal | 0.0 | 11.5 | - |
> | Qwen2.5 VL | 3B | Multimodal | 10.1 | 18.7 | - |
> | Qwen2.5 VL | 4x3B | Multimodal | 12.6 | 21.3 | - |
> | DigiRL (Qwen2.5 VL) | 3B | Multimodal | 22.3 | 35.2 | 16.5 |
> | DigiRL (Qwen2.5 VL) | 4x3B | Multimodal | 20.1  | 38.7  | 20.0 |
> | UIAgent (Qwen2) | 7B | Text | 18.9  | 48.4  | 35.5 |
> | UIAgent (Qwen2) | 4x7B | Text | 21.4  | 53.2  | 40.3 |
> | UIAgent (Qwen2) | 6x7B | Text | 18.9  | 54.3  | 41.5 |
> | CollabUIAgents (Qwen2) | 4x7B | Text | 29.3  | **61.2**  | **48.3** |
> | CollabUIAgents (Qwen2) | 6x7B | Text | **32.7**  | 59.7  | 46.9 |
>
> These results highlight that the performance advantage of CollabUIAgents stem from its unique MARL framework that effectively learns and fosters complex collaboration among agents.
>
> Hope for your continued valuable insights and favorable consideration!

---

> ### Author Response · Authors · 2025-06-09
>
> Dear Reviewer sVoJ,
>
> We would like to express our sincere gratitude for your time and the thoughtful feedback you have provided.
>
> We look forward to your comments on our responses and would be delighted to address any further questions or concerns you might have.
>
> Thank you again for your valuable insights and for taking the time to review our work.
>
> Sincerely,
>
> The Authors

---

> > ### Comment · Reviewer_sVoJ · 2025-06-10
> >
> > Thanks for the response, I will maintain my score.

---

> > > ### Author Response · Authors · 2025-06-10
> > >
> > > Thank you for your valuable feedback and comments. We have carefully addressed your comments in our revision.
> > >
> > > With the rebuttal deadline approaching, we would be pleased to address any further questions and provide any additional clarifications. Your feedback has been invaluable in strengthening our manuscript.
> > >
> > > Sincerely,
> > >
> > > The Authors

---

### Official Review · Reviewer_wGPK · 2025-05-12

**Rating:** 6
**Confidence:** 3
**Ethics Flag:** 1

**Summary:**

The paper presents a MARL (multi-agent RL) approach for an environment with sparse rewards and where an LLM is able to provide critic feedback. Empirical results show that the multi-agent system does well compared to strong single agent systems and has an ability to generalise.

**Questions To Authors:**

Were the number of agents experimented with? It would be interesting to see results on different numbers of agents, rather than just the n=4 and m=3 setting. How robust is this? Why was this specific setting chosen?

**Reasons To Accept:**

The proposed system clearly improves upon the single agent systems, as demonstrated by results on public benchmarks reported in tables 1 and 2.
The ability to introduce new LLM agents is a good property.
The structure of the paper is good, and the takeaways outline clearly the claims of the paper nicely.

**Reasons To Reject:**

It's not clear (or I have missed) important details about what LLM is used for the reward model.

There is no comparison against multi-agent systems. The proposed MARL system is only compared against strong but single agent systems. I don't know the MARL literature well, but I believe a comparison of multi-v-single is misleading, and a multi-v-multi comparison would help demonstrate whether this proposed multi agent approach is an incremental improvement.

There's a lack of detail on how the environment provides the binary feedback. Says "based on static rules" on line 127. Presumably this is information available in the public benchmarks themselves? It would be helpful to note this if so, and/or say a little more about what success means in the environment and how this is measured.

---

> ### Author Response · Authors · 2025-06-01
> **(1/n) Responses by Authors**
>
> Thanks for your careful and insightful reviews.
>
> > `Weakness 1`: It's not clear important details about what LLM is used for the reward model.
>
> Thank you for your valuable feedback regarding the clarity of the LLM used for the credit re-assignment (CR) mechanism.  In our CollabUIAgents framework, the critic agent, which performs the credit re-assignment during the multi-agent reinforcement learning (MARL) phase, utilizes GPT-4o-2024-08-06. This is stated in **Appendix C.3.2** CollabUIAgents Framework. To further enhance clarity and ensure this crucial detail is more accessible, **we will revise the manuscript and explicitly name the LLM (GPT-4o-2024-08-06) used for the critic agent when it is first introduced in Section 3.2 Language Multi-Agent RL**.
>
>
> > `Weakness 2`: There is no comparison against multi-agent systems.
>
> Thank you very much for your valuable feedback, especially your suggestion regarding the comparison with other multi-agent systems to more comprehensively evaluate our proposed MARL framework. We acknowledge the importance of 'multi-agent vs. multi-agent' comparisons in evaluating MARL systems. **In our current manuscript, we have conducted some multi-agent comparisons**:
> - **Table 1** includes a comparison with the multi-agent version (4 agents) of DigiRL. The experimental results show that our 'CollabUIAgents_mobile' (4 agents) outperforms this 4-agent DigiRL system in both AndroidWorld and MMiniWoB++ (e.g., SR of 29.3 vs. 20.1 in AndroidWorld, and SR of 61.2 vs. 38.7 in MMiniWoB++).
> - Furthermore, **the ablation study in Table 3** also indirectly provides a perspective on multi-agent comparisons. For instance, it shows that 'CollabUIAgentsmobile' (n=4) surpasses a 'Base UIAgent' (n=4) system that also uses 4 agents but without our CR strategy and MARL optimization. This highlights the effectiveness of our CR strategy and MARL framework.
>
> **In response to your concerns, we have conducted additional experiments** and expanded the original Table 1 to provide a more comprehensive perspective on multi-agent comparisons. Our updated experimental results table (extended Table 1) is shown below, which includes comparisons against a Multi-Agent Reinforcement Learning (MARL) baseline as well as various "naive multi-agent ensemble" configurations:
>
> | Method | Params  | Input  |   SR(Androidworld)  | SR(MMiniWoB++)  | ∆ Generalization  |
> |-------------|----------|--------------|----------| ----------| ----------|
> | **Open-Scoure** |
> | Qwen2 | 7B | Text |  6.2 | 12.9  | - |
> | Qwen2 | 4x7B | Text |  4.3 | 15.2  | - |
> | Qwen2 VL | 2B | Multimodal | 0.0 | 10.0 | - |
> | Qwen2 VL | 4x2B | Multimodal | 0.0 | 11.5 | - |
> | Qwen2.5 VL | 3B | Multimodal | 10.1 | 18.7 | - |
> | Qwen2.5 VL | 4x3B | Multimodal | 12.6 | 21.3 | - |
> | DigiRL (Qwen2.5 VL) | 3B | Multimodal | 22.3 | 35.2 | 16.5 |
> | DigiRL (Qwen2.5 VL) | 4x3B | Multimodal | 20.1  | 38.7  | 20.0 |
> | UIAgent (Qwen2) | 7B | Text | 18.9  | 48.4  | 35.5 |
> | UIAgent (Qwen2) | 4x7B | Text | 21.4  | 53.2  | 40.3 |
> | UIAgent (Qwen2) | 6x7B | Text | 18.9  | 54.3  | 41.5 |
> | CollabUIAgents (Qwen2) | 4x7B | Text | 29.3  | **61.2**  | **48.3** |
> | CollabUIAgents (Qwen2) | 6x7B | Text | **32.7**  | 59.7  | 46.9 |
>
> **Comparison with DigiRL (Qwen2.5 VL)** (4 agents), Our CollabUIAgents achieved Success Rates (SR) of 29.3% on AndroidWorld and 61.2% on MMiniWoB++. **Comparison with naive multi-agent ensembles**, CollabUIAgents (4x7B) significantly outperforms the UIAgent (Qwen2) (4x7B) ensemble. Similarly, CollabUIAgents (6x7B) (SR 32.7%, 59.7%) significantly outperforms the UIAgent (Qwen2) (6x7B) ensemble. These results highlight that the performance advantage of CollabUIAgents does not solely stem from using more model parameters or multiple decision units, but rather from its unique MARL framework that effectively learns and fosters complex collaboration among agents.

---

> ### Author Response · Authors · 2025-06-01
> **(2/n) Responses by Authors**
>
> > `Weakness 3`: There's a lack of detail on how the environment provides the binary feedback.
>
> Thank you for your insightful comment regarding the need for more detail on how the environment provides binary feedback and defines task success. You are correct in presuming that the 'static rules' mentioned on line 127  for determining the binary **outcome reward are indeed predefined by the public benchmark environments**. In these benchmarks, success is typically defined as the agent's successful completion of the specific task instruction provided by the environment. **For instance**, in AndroidWorld, which features programmatic tasks, success involves the agent correctly executing a sequence of UI operations to achieve a predefined goal (e.g., 'send an email to a specific contact,' 'set an alarm for a particular time,' or 'move a file from one folder to another'). To enhance the clarity of our manuscript, **we will explicitly state that these 'static rules' and the definition of the successful terminal state are inherent to the specific tasks within the public benchmarks employed**. We believe these additions will adequately address your concerns and make the process of how success is measured and binary feedback is provided by the environments much clearer.
>
>
> > `Question`:  Were the number of agents experimented with? It would be interesting to see results on different numbers of agents, rather than just the n=4 and m=3 setting. How robust is this? Why was this specific setting chosen?
>
> Thank you for your question. **We have conducted additional experiments to address this**, with results included in an expanded Table 1. we experimented with varying numbers of agents. The updated Table 1 (excerpt below) shows performance for CollabUIAgents (Qwen2) with n=4 and n=6 agents, and for UIAgent (Qwen2) (our base agent, serving as a naive ensemble baseline) with n=1, n=4, and n=6 agents. The number of conversation rounds, m, was kept at 3 for these comparisons.
>
> | Method | Params  | Input  |   SR(Androidworld)  | SR(MMiniWoB++)  | ∆ Generalization  |
> |-------------|----------|--------------|----------| ----------| ----------|
> | UIAgent (Qwen2) | 7B | Text | 18.9  | 48.4  | 35.5 |
> | UIAgent (Qwen2) | 4x7B | Text | 21.4  | 53.2  | 40.3 |
> | UIAgent (Qwen2) | 6x7B | Text | 18.9  | 54.3  | 41.5 |
> | CollabUIAgents (Qwen2) | 4x7B | Text | 29.3  | **61.2**  | **48.3** |
> | CollabUIAgents (Qwen2) | 6x7B | Text | **32.7**  | 59.7  | 46.9 |
>
> **For CollabUIAgents (Qwen2)**: Increasing agents from n=4 to n=6 improved performance on the training environment (AndroidWorld SR: 29.3% → 32.7%). However, on the unseen generalization environment (MMiniWoB++ SR: 61.2% → 59.7%), performance slightly decreased, as did the ∆Generalization. This indicates that while more agents can enhance performance on trained tasks, there might be a trade-off affecting generalization, possibly due to increased complexity or optimization challenges.
>
> **For UIAgent (Qwen2)**: Performance trends were less consistent when increasing agent numbers, with n=6 showing a drop in AndroidWorld performance compared to n=4. CollabUIAgents at both n=4 and n=6 significantly outperformed the corresponding UIAgent naive ensemble configurations, highlighting the effectiveness of our MARL framework and CR strategy.
>
> **Regarding to why was this specific setting chosen**: Although our framework's edge update mechanism allows for dynamic adaptation of the communication structure, potentially leading to an implicit optimization of active agent numbers, computational constraints led us to explicitly test fixed settings like n=4, which proved highly effective and practical. The choice of m=3 conversation rounds aimed to ensure sufficient agent interaction for collaborative decision-making  without incurring excessive latency or overly long contexts. The n=4, m=3 configuration for CollabUIAgents was chosen based on a balance of empirical performance, generalization capability, and computational considerations. Specifically, the n=4 setting demonstrated superior generalization compared to n=6 in key tests, while robustly outperforming single-agent, naive ensemble, and existing MARL baselines.

---

> ### Author Response · Authors · 2025-06-08
> **(3/n) Responses by Authors**
>
> > `Comparison with distillation baseline`
>
> Thank you for your review and time. To further validate the effectiveness of our approach, **we introduce 'Distilling' baselines**, which were train via supervised fine-tuning (SFT) directly on data where GPT-4 provided the correct answers (i.e., successful trajectories/actions). We ensure that the volume of data used for this SFT-based distillation was consistent with the amount of preference data used for DPO. The results from the expanded Table 1 are as follows:
>
> | Method | Params  | Input  |   SR(Androidworld)  | SR(MMiniWoB++)  | ∆ Generalization  |
> |-------------|----------|--------------|----------| ----------| ----------|
> | **Closed-Scoure** |
> | M3A (GPT4) | N/A | Text | **30.6** | 59.7 | - |
> | M3A (GPT4) | N/A | Multimodal | 25.4 | **67.7** | - |
> | **Open-Scoure** |
> | Qwen2 | 7B | Text |  6.2 | 12.9  | - |
> | Qwen2 VL | 2B | Multimodal | 0.0 | 10.0 | - |
> | Distilling (Qwen2) | 7B | Text | 12.3 | 35.8 | 22.9 |
> | Distilling (Qwen2) | 4x7B | Text | 15.2 | 38.0 | 25.1 |
> | UIAgent (Qwen2) | 7B | Text | 18.9  | 48.4  | 35.5 |
> | UIAgent (Qwen2) | 4x7B | Text | 21.4  | 53.2  | 40.3 |
> | CollabUIAgents (Qwen2) | 4x7B | Text | **29.3**  | **61.2**  | **48.3** |
>
> The Distilling (Qwen2) 7B model, trained via this SFT approach on GPT-4's correct outputs, achieved SRs of only 13.3% and 35.8%. Even the Distilling (Qwen2) 4x7B model only reached SRs of 15.2% and 38.0%. **These results strongly indicate that the performance achieved by our CollabUIAgents framework, through its novel MARL with CR strategy**, significantly surpasses methods based on direct SFT distillation from a strong teacher model like GPT-4.
>
> Hope for your continued valuable insights and favorable consideration!

---

> ### Author Response · Authors · 2025-06-09
>
> Dear Reviewer wGPK,
>
> We are truly grateful for your time and the thoughtful feedback you have provided.
>
> We look forward to your comments on our responses and would be delighted to address any further questions or concerns you might have.
>
> Thank you again for your valuable insights and for taking the time to review our work.
>
> Sincerely,
>
> The Authors

---

### Official Review · Reviewer_9rgY · 2025-05-14

**Rating:** 6
**Confidence:** 3
**Ethics Flag:** 1

**Summary:**

This paper explores learning in multi-agent systems using small language models, aiming to achieve strong generalization performance without relying on predefined roles for each agent. Specifically, the authors apply MARL, where each agent has its own policy, and introduce a critic agent that assigns rewards and updates value estimates based on the rollout of actions. The experiments test whether this approach can enable UI dialogue in diverse environments, and the authors report that their 7B-model achieves performance comparable to much larger open models.

**Questions To Authors:**

I found the performance gain in the CollabUIAgents setting (mobile to m→web) particularly striking regarding the experimental results. While this is presumably an intended outcome, it would be helpful if the authors could provide more analysis or discussion of which types of cases benefited most from the proposed approach.

The case studies in the Appendix are informative. However, please indicate which model/configuration each case corresponds to. It would also strengthen the paper to include case studies demonstrating generalization, if such results are claimed.

**Reasons To Accept:**

The work presents a highly interesting attempt to improve a known weakness in multi-agent systems, the difficulty of generalizing beyond the specialized behavior of individual agents.

The experimental comparisons seem reasonable when the paper is read as a proposal for applying MARL to language-based agents. The experiments themselves are comprehensive.

**Reasons To Reject:**

There is room to improve clarity in how the core ideas of the proposed method relate to the specific problems being addressed. Methodologically, the work could be interpreted as an extension of reward/credit assignment approaches in multi-agent learning (e.g., Qu et al., 2025; Lin et al., 2025). However, as a whole, the paper proposes a solution tailored explicitly to applying MARL to language agents. I suggest providing a more transparent and precise explanation of the scope of the paper’s contribution in this regard.

Tables 1 and 2 are complex and challenging to interpret, as they attempt to compare a wide range of experimental conditions. While space limitations are understandable, I recommend streamlining the tables to focus more clearly on key comparisons relevant to the central points of discussion.

---

> ### Author Response · Authors · 2025-06-01
> **(1/n) Responses by Authors**
>
> Thanks for your careful and insightful reviews.
>
> > `Weakness 1`: There is room to improve clarity in how the core ideas of the proposed method relate to the specific problems being addressed.
>
> Response: To more precisely clarify the distinctions of our work (CollabUIAgents) compared to Qu et al. (2025) and Lin et al. (2025), especially regarding its contributions to **MARL applications for language agents in interactive environments**, we will include the following concise comparison table in the revised manuscript:
>
> | Feature | CollabUIAgents  | Lin et al., 2025 (LCA)  |   Qu et al., 2025 (LaRe)  |
> |-------------|----------|--------------|----------|
> | Primary Focus |  Performance & generalization of language multi-agent systems in interactive UI environments  |  MARL credit assignment with sparse team rewards | Episodic RL credit assignment; tackles redundancy & ambiguity |
> | Core Application | Application of language agents in interactive UI environments.  | MARL agents & environments (Grid World, Pistonball)  | RL agents & environments (MuJoCo, MPE)  |
> | Generated Reward Mechanism | Fine-grained process rewards used to synthesize preference data for DPO | Dense, agent-specific, potential-based rewards trained from LLM rankings | Proxy rewards derived from LLM-code-generated latent reward encoder & decoder  |
> | Focus on Generalization | ✅ | ❌ | ❌  |
> | For Interactive environment | ✅ | ❌ | ❌  |
> | Dynamic MARL Communication | ✅ | ❌ | ❌  |
>
> As the table indicates, while Lin et al. (LCA)  and Qu et al. (LaRe)  also explore LLMs for credit assignment, their primary focus is on addressing sparse rewards in general MARL  or latent reward encoding in episodic RL, respectively. In contrast, the core contributions and distinctions of CollabUIAgents are:
> - **Specifically Designed for Language Agents & Interactive UI Environments**:  Our framework focuses on solving performance and generalization challenges for language agents operating in UI environments like mobile navigation and web Browse, which present complex observation and action spaces.
> - **Process Reward Mechanism via LLM Understanding**:  We innovatively employ an LLM as a Critic Agent to assign fine-grained "process rewards". This evaluates the contribution of each decision step within a task sequence based on the LLM's comprehension, rather than relying solely on sparse environmental signals or state ranking. These process rewards are then used to synthesize preference data for policy learning via DPO.
> - **Emphasis on Cross-Environment Generalization**:  Our Credit Re-assignment (CR) strategy is designed to leverage the general world knowledge of LLMs. Combined with continual MARL  and mechanisms like communication structure (edge) updates, this significantly enhances the generalization capabilities of language multi-agent systems in unseen environments.
>
> We believe these revisions more effectively underscore our work's unique positioning and innovations. Thank you for your feedback.

---

> ### Author Response · Authors · 2025-06-01
> **(2/n) Responses by Authors**
>
> > `Weakness 2`: Tables 1 and 2 are complex and challenging to interpret.
>
> Response:  Thank you for your valuable feedback regarding the clarity of the tables. We agree that Tables 1 and 2 currently include many experimental conditions, which may make them complex to interpret. To address this issue and enable readers to focus more clearly on the core strengths of our work, we plan to streamline these two tables.
>
>
> | Method | Params  | Input  |   SR(Androidworld)  | SR(MMiniWoB++)  | ∆ Generalization  |
> |-------------|----------|--------------|----------| ----------| ----------|
> | **Closed-Scoure** |
> | M3A (GPT4) | N/A | Text | **30.6** | 59.7 | - |
> | M3A (GPT4) | N/A | Multimodal | 25.4 | **67.7** | - |
> | M3A (Gemini1.5) | N/A | Text | 19.4 | 57.4 | - |
> | M3A (Gemini1.5) | N/A | Multimodal | 22.8 | 40.3 | - |
> | SeeAct (GPT4) | N/A | Multimodal | 15.5 | 66.1 | - |
> | **Open-Scoure** |
> | Qwen2 | 7B | Text |  6.2 | 12.9  | - |
> | Qwen2 VL | 2B | Multimodal | 0.0 | 10.0 | - |
> | Qwen2.5 VL | 3B | Multimodal | 10.1 | 18.7 | - |
> | InfiGUIAgent (Qwen2 VL) | 2B | Multimodal | 9.1 | 15.6 | 5.6 |
> | DigiRL (Qwen2.5 VL) | 3B | Multimodal | 22.3 | 35.2 | 16.5 |
> | DigiRL (Qwen2.5 VL) | 4x3B | Multimodal | 20.1  | 38.7  | 20.0 |
> | UIAgent (Qwen2) | 7B | Text | 18.9  | 48.4  | 35.5 |
> | UIAgent (Qwen2) | 4x7B | Text | 21.4  | 53.2  | 40.3 |
> | CollabUIAgents (Qwen2) | 4x7B | Text | **29.3**  | **61.2**  | **48.3** |
>
> **Table 1**: Experimental results on mobile operation environments. Success rates (SR) in
> AndoridWorld and MobileMiniWoB++ (MMiniWoB++) are listed. ∆ Generalization indicates
> the performance gap between the base model and agent learning methods based on the
> model in MobileMiniWoB++.
>
> | Method | Params  | Input  |   SSR(Mind2Web)  | SR(AutoWebBench)  | ∆ Generalization  |
> |-------------|----------|--------------|----------| ----------| ----------|
> | **Closed-Scoure** |
> | GPT-3.5 Turbo | N/A | Text |  17.4 | 10.7  | - |
> | GPT-4 | N/A | Text | 30.9  | 37.8  | - |
> | Claude2 | N/A | Text | -  | 10.5 | - |
> | **Open-Scoure** |
> | Qwen2 | 7B | Text |  7.4  | 8.5   | - |
> | LLaMA2 | 7B | Text |  - | 2.9  | - |
> | LLaMA2 | 70B | Text |  -  | 10.6  | - |
> | SFT (Qwen2 VL) | 9.6B | Multimodal |  10.1  | -  | - |
> | SFT (SeeClick) | 9.6B | Multimodal |  20.9  | -  | - |
> | UIAgent (Qwen2) | 7B | Text | 11.9  | 12.8  | 4.3 |
> | UIAgent (Qwen2) | 4x7B | Text | 13.2  |  14.0 | 5.5 |
> | CollabUIAgents_mobile (Qwen2) | 4x7B | Text |  16.2 |  17.7  | 9.2 |
> | CollabUIAgents_m_web (Qwen2) | 4x7B | Text |  **30.7** | **34.7**  | **26.2** |
>
> **Table 2**: Experimental results on web browsing environments. Average step success rates
> (SSR) in Mind2Web and AutoWebBench are reported. “SFT” denotes the base model is
> supervised fine-tuned in Mind2Web. ∆Generalization indicates the gap between the base
> model and agent learning methods based on the model in AutoWebBench.
>
>
> > `Question 1`:  It would be helpful if the authors could provide more analysis or discussion of which types of cases benefited most from the proposed approach.
>
> The performance improvement you observed is indeed a key outcome we aimed to achieve. This significant enhancement primarily stems from several core designs of our CollabUIAgents framework, especially when combined with Continual Multi-Agent Reinforcement Learning (Continual MARL) for cross-environment adaptation:
> - **Generalization and Transfer of Foundational Knowledge**: CollabUIAgentsmobile first learns general UI interaction patterns and collaborative behaviors in the mobile environment through our credit re-assignment (CR) strategy. This knowledge, such as identifying buttons, inputting text, and understanding basic navigation logic, is largely transferable to web environments, as many interaction elements and task flows are conceptually similar.
> - **Efficient Adaptation in a New Domain via Continual MARL and the CR Strategy**: Learning with preference data synthesized by an adversarial agent allows the policy to optimize efficiently from a comparison of "correct" versus "incorrect" actions. When agents enter the new web environment, the LLM-guided CR can provide effective learning signals based on a universal understanding of web interactions, even if these interaction details differ from the mobile environment. This allows agents to grasp the "rules" of web tasks more quickly. Continual learning enables agents to adjust their policies for elements unique to web environments (e.g., HTML structure, specific webpage layouts, and navigation patterns).
>
> **Regarding to Which Types of Cases Benefited Most**: based on the analysis above, and according to our statistics, Navigation and Operation Tasks benefit most. These tasks share high-level logic (e.g., search, select, fill, submit), but specific UI elements, layouts, and steps can vary by website. The CR strategy and Continual MARL help agents quickly adapt to these differences while leveraging existing high-level task understanding.

---

> ### Author Response · Authors · 2025-06-01
> **(3/n) Responses by Authors**
>
> > `Question 2`:  The case studies in the Appendix are informative. However, please indicate which model/configuration each case corresponds to.
>
> Thank you very much for your valuable suggestions. This case study in Appendix A aims to demonstrate the specific steps taken by our proposed CollabUIAgents mobile system (**trained in the AndroidWorld environment, using Qwen2 7B as the base model, with a configuration of 4 agents, each engaging in 3 rounds of conversation**) during task execution, and how the credit re-assignment (CR) process correctly assigns rewards to each action. **We will explicitly state in the caption of Figure 4 that this case corresponds to the CollabUIAgents mobile system and its key configurations**.

---

> ### Author Response · Authors · 2025-06-08
> **(4/n) Responses by Authors**
>
> We are truly grateful for your thorough review and constructive feedback throughout this process. **Your insightful comments have been invaluable in helping us to significantly improve the quality and clarity of our manuscript!**
>
> We remain committed to addressing all feedback to ensure the final version of our paper is as strong as possible and **have specifically conducted additional experiments exploring different numbers of agents to demonstrate the robustness and generalizability of our method**. Our updated experimental results table (extended Table 1) is shown below:
>
> | Method | Params  | Input  |   SR(Androidworld)  | SR(MMiniWoB++)  | ∆ Generalization  |
> |-------------|----------|--------------|----------| ----------| ----------|
> | **Open-Scoure** |
> | Qwen2 | 7B | Text |  6.2 | 12.9  | - |
> | Qwen2 | 4x7B | Text |  4.3 | 15.2  | - |
> | Qwen2 VL | 2B | Multimodal | 0.0 | 10.0 | - |
> | Qwen2 VL | 4x2B | Multimodal | 0.0 | 11.5 | - |
> | Qwen2.5 VL | 3B | Multimodal | 10.1 | 18.7 | - |
> | Qwen2.5 VL | 4x3B | Multimodal | 12.6 | 21.3 | - |
> | DigiRL (Qwen2.5 VL) | 3B | Multimodal | 22.3 | 35.2 | 16.5 |
> | DigiRL (Qwen2.5 VL) | 4x3B | Multimodal | 20.1  | 38.7  | 20.0 |
> | UIAgent (Qwen2) | 7B | Text | 18.9  | 48.4  | 35.5 |
> | UIAgent (Qwen2) | 4x7B | Text | 21.4  | 53.2  | 40.3 |
> | UIAgent (Qwen2) | 6x7B | Text | 18.9  | 54.3  | 41.5 |
> | CollabUIAgents (Qwen2) | 4x7B | Text | 29.3  | **61.2**  | **48.3** |
> | CollabUIAgents (Qwen2) | 6x7B | Text | **32.7**  | 59.7  | 46.9 |
>
> These results highlight that the performance advantage of CollabUIAgents stem from its unique MARL framework that effectively learns and fosters complex collaboration among agents.
>
> Hope for your continued valuable insights and favorable consideration!

---

> ### Author Response · Authors · 2025-06-09
>
> Dear Reviewer 9rgY,
>
> We sincerely appreciate the time and effort you have invested in offering us your valuable feedback.
>
> We look forward to your comments on our responses and would be delighted to address any further questions or concerns you might have.
>
> Thank you once more for your insightful comments and the time you have dedicated to our submission.
>
> Best,
>
> The Authors

---

> ### Comment · Reviewer_9rgY · 2025-06-10
> **reply to author's comments**
>
> Thank you for the detailed response.
> I am reasonably confident in the technical contribution of this paper; however, there was room for improvement in how that contribution was presented.
> The additional comparison tables and the reorganization of experimental results are expected to improve the presentation to some extent.
> Since the revised manuscript is not available during the rebuttal phase, I believe this improvement is necessarily limited.
> While I remain positive about the paper, my score also depends on the support of other reviewers.
> I prefer to maintain my current score.

---

> > ### Author Response · Authors · 2025-06-11
> >
> > We have incorporated additional comparison data and reorganized the experimental section as suggested, and we believe the revised manuscript will more clearly demonstrate the research contributions. Thank you for recognizing the value of our work and for your constructive feedback.
> >
> > Sincerely,
> > The Authors

---

### Author Response · Authors · 2025-06-09
**General Response by Authors**

We would like to express our sincere appreciation to the reviewers for the time and effort they have dedicated to our paper. Their insightful feedback and constructive suggestions have been invaluable in enhancing the quality of our manuscript. We are truly encouraged and grateful for the reviewers' recognition of several key aspects of our work:

- The work pushes in an **extremely important direction** (Reviewer Mpaq) **by addressing a known weakness in multi-agent systems** (Reviewer 9rgY). The application of MARL to mobile/web automation is a **relatively under-explored area with practical value** (Reviewer sVoJ).
- The work presents a **solid proof of concept for many good ideas** (Reviewer Mpaq). Specifically, the Credit Re-assignment (CR) strategy was highlighted as a contribution that could **potentially benefit other cooperative multi-agent scenarios** (Reviewer sVoJ).
- The **experiments are comprehensive** (Reviewer 9rgY) and that our system **clearly improves upon the single agent systems** and demonstrates the **ability of generalization**. (Reviewer wGPK).
- The paper's **structure is good**, and the **takeaways outline clearly the claims** of the paper nicely (Reviewer wGPK). The core concept is **convincing and well-presented** in the paper (Reviewer Mpaq).


We have carefully considered and responded to all the reviewers' comments and have incorporated the necessary revisions into the manuscript. We sincerely hope our responses and the updated manuscript address all their concerns.

---

### Decision · Program_Chairs · 2025-07-08

**Decision:**

Accept

**Comment:**

While multi-agent systems outperform single agents in interactive tasks (e.g., web browsing), their generalization across environments remains limited by rigid role definitions. This paper proposes CollabUIAgents, a reinforcement learning framework that uses LLM-guided credit re-assignment (CR) and synthetic preference data to develop role-free, generalizable policies, and achieves performance comparable to closed-source 7B models. The reviewers provided valuable feedback, which the authors have made great efforts to address. One reviewer maintained their original low score without providing specific justification following the rebuttal, so I have discounted this evaluation in my assessment. Given the authors' rebuttal and other reviewers' supportive scores, I recommend acceptance.